# DICTFORMER: TINY TRANSFORMER WITH SHARED DICTIONARY

**Qian Lou, Ting Hua, Yen-Chang Hsu, Yilin Shen, Hongxia Jin**
Samsung Research America
{qian.lou, ting.hua, yenchang.hsu, yilin.shen, hongxia.jin}@samsung.com

## ABSTRACT

We introduce DictFormer with efficient shared dictionary to provide a compact, fast, and accurate transformer model. DictFormer significantly reduces the redundancy in the transformer's parameters by replacing the prior transformer's parameters with compact, shared dictionary, few unshared coefficients and indices. Also, DictFormer enables faster computations since expensive weights multiplications are converted into cheap shared look-ups on dictionary and few linear projections. Training dictionary and coefficients are not trivial since indices used for looking up dictionary are not differentiable. We adopt a sparse-constraint training with $l_1$ $norm$ relaxation to learn coefficients and indices in DictFormer. DictFormer is flexible to support different model sizes by dynamically changing dictionary size. Compared to existing lightweight Transformers, DictFormer consistently reduces model size over Transformer on multiple tasks, e.g., machine translation, abstractive summarization, and language modeling. Extensive experiments show that DictFormer reduces $3.6\times$ to $8.9\times$ model size with similar accuracy over multiple tasks, compared to Transformer.

## 1 INTRODUCTION

Transformer (Vaswani et al., 2017) has been widely used in natural language processing (NLP) for its superior capability in capturing long-distance dependencies. However, its good performance comes with the giant model size. For example, T5 (Raffel et al., 2019) with a hidden dimension of 65K and GPT-3 (Brown et al., 2020) with 96 transformer blocks have 11 billion and 175 billion parameters, respectively. These large Transformers suffer from multiple severe issues, such as complicated learning and difficult deployment on mobile/IoT devices. First, during the training of a big transformer model, large training corpora (Raffel et al., 2019; Brown et al., 2020) or careful regularization (Merity et al., 2017; Mehta et al., 2021) are required. Furthermore, the trained model is over-parameterized (Reid et al., 2021; Zhao et al., 2021). The large model size with 11 billion and 175 billion parameters is beyond the capabilities of many edge devices including mobile devices and IoTs. Therefore, there is an urgent need to design parameter-efficient and fast transformer model that eliminates redundant parameters and enables real-time NLP applications on the edge.

Weights sharing is a choice proved by ALBERT (Lan et al., 2020) for designing compact and efficient pre-trained transformer encoders, like BERT, on self-supervised learning task. However, directly sharing all weights in an encoder-decoder transformer for a sequence-to-sequence task like machine translation will dramatically decrease performance and accuracy (Reid et al., 2021). Although a recent framework Universal Transformer (UT) (Dehghani et al., 2019) shows that a vanilla transformer's accuracy can be improved by using recursive weight sharing, UT's high accuracy comes at the cost of deeper blocks or wider hidden dimensions, which significantly enlarges the computational cost and does not necessarily reduce the model size.

This paper introduces a new compact, fast, and accurate transformer architecture, DictFormer, that can be easily trained and deployed on edge devices. DictFormer depends on dictionary sharing and unshared linear projection coefficients instead of weights sharing. Specifically, a shared dictionary among all encoder/decoder blocks can significantly reduce the parameter redundancy and therefore compresses the model size. Few unshared linear projection with coefficients on the shared dictionary enable each encoder/decoder block to have distinct layer-wise feature representations,

thus improving the representation abilities compared to prior weights sharing. Also, DictFormer provides a method to dynamically control the layer-wise representation abilities by using the group-wise shared dictionary for layers with large-dimension features, e.g., Feed-forward Network (FFN). Last but not least, we show that training dictionary and coefficients are not trivial since indices used for looking up dictionary are not differentiable. We convert coefficients and indices into a sparse matrix and train it with $l_1\ norm$ relaxation and convert this sparse matrix into dense coefficients and indices during inference.

Extensive experiments demonstrate that our DictFormer provides significant improvements over existing transformers on three sequence-to-sequence tasks, (i) machine translation, (ii) abstractive summarization, and (iii) language modeling. For machine translation, on IWSLT 2014 German-English, DictFormer attains transformer performance with $8.9\times$ fewer parameters and $2\times$ fewer Multi-Adds; on WMT 2014 German-English, DictFormer brings about $4.9\times$ model compression and $1.9\times$ computation reduction; on WMT 2014 English-French, DictFormer obtains consistent performance improvements: $4.9\times$ model compression and $1.9\times$ less computation with a similar score. For abstractive summarization, DictFormer reduces the model size by more than $4.7\times$ on CNN-DailyMail dataset. For language modeling, DictFormer matches the performance of transformer with $3.6\times$ to $5.7\times$ fewer parameters than transformer on WikiText-103 benchmark.

## 2 RELATED WORK AND MOTIVATION

**Related lightweight Transformers.** Several methods have been proposed to design lightweight transformers. The first line of research is to reduce the transformer computation complexities by redesigning self-attention mechanism including (Katharopoulos et al., 2020; Zhou et al., 2021; Raganato et al., 2020; You et al., 2020; Correia et al., 2019; Kaiser et al., 2018). These methods cannot reduce model size. The second line of research is model compression, e.g., quantization (Prato et al., 2020; Lou et al., 2020b), pruning (Behnke & Heafield, 2020), low-rank factorization (Ma et al., 2019), and knowledge distillation (Wang et al., 2020). These two directions of research can be combined with our Dictformer. The third line of research is efficient architecture design (So et al., 2019; Wu et al., 2020; Mehta et al., 2021) by improving the expressiveness of transformers. The forth line of research is weights sharing (Xia et al., 2019; Ma et al., 2019; Dehghani et al., 2019; Reid et al., 2021; Takase & Kiyono, 2021) by reusing parameters across transformer blocks. Weights sharing cannot reduce computations. Our Dictformer falls into the third and forth category. We show that our Dictformer with Dictionary sharing can reduce both model size and computations.

**Transformers with Weights Sharing.** Weight sharing is surprisingly effective to compress model size for discriminate NLP models based on Transformer encoders, e.g., BERT. For example, prior work ALBERT (Lan et al., 2020) shows that even sharing all parameters across layers does not introduce any accuracy reduction. However, for generative sequence-to-sequence models based on transformer's encoders and decoders, sharing all parameters will significantly decrease accuracy on multiple standard machine translation or language modelling tasks (Reid et al., 2021; Takase & Kiyono, 2021). To match vanilla Transformer's accuracy, multiple works (Reid et al., 2021; Takase & Kiyono, 2021; Xia et al., 2019; Ma et al., 2019) only share weights across partial layers instead of all layers. However, partial weights sharing remarkably brings down the model size compression effect of weights sharing. Also, how to decide which layers should be shared in partial weights sharing is difficult due to the large and dynamic search space that is dependent on the specific tasks.

Transformer with all-parameters sharing such as Universal Transformer (Dehghani et al., 2019) matches or improves transformer's performance at the cost of a wider or deeper transformer architecture. A wider transformer with a larger embedding dimension enlarges the model size and brings larger computations (Mult-Adds). A deeper transformer with more encoder/decoder blocks does not only increase model size, but also introduces more computations. Importantly, weights sharing techniques cannot reduce Mult-Adds numbers and training/inference time. Figure 6(a) in Appendix shows the comparisons of Transformers with weights sharing and our dictionary sharing. Weights sharing techniques cannot solve the deployment challenges of transformers on resource-limited devices for real-time NLP applications.

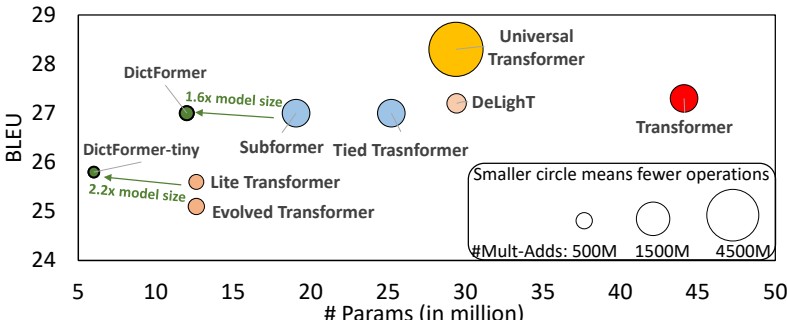

Figure 1: DictFormer achieves similar BLEU score with prior works using fewer parameters and Mult-Adds on WMT 2014 De-En translation. DictFormer is flexible and scalable, e.g., for mobile devices, DictFormer-tiny can reduce 2.2× model size of lite transformer (Wu et al., 2020) that achieves state-of-the-art performance under mobile settings. Results of existing works come from their own implementations and #Params does not include embedding parameters, *e.g.*, Transformer (Vaswani et al., 2017) has 44M #Params.

**Motivation of DictFormer with Dictionary Sharing.**

Figure 1 shows that although transformers with weights sharing including Universal Transformer (Dehghani et al., 2019), Tied Transformer (Xia et al., 2019), and Subformer (Reid et al., 2021) significantly reduce the model size over vanilla transformer and lightweight transformer DeLighT (Mehta et al., 2021), they cannot reduce the #Mult-Adds and even suffer from larger #Mult-Adds. Particularly, Tied Transformer (Xia et al., 2019) and Subformer (Reid et al., 2021) compress transformer model by $1.7\times \sim 2.6\times$ but cannot reduce #Mult-Adds. Universal Transformer (Dehghani et al., 2019) achieves $\sim 1$ BLEU score improvement with $1.4\times$ less parameters, but the #Mult-Adds is increased by $\sim 4.3\times$, thereby significantly prolonging the NLP inference latency and restraining the deployment of real-time NLP applications on edge devices. To enable the deployment of transformer models on mobile devices, recent work Evolved Transformer (So et al., 2019) and Lite Transformer (Wu et al., 2020) try to design new lightweight transformer architecture to meet the defined mobile settings, e.g., 10 million parameters, but their tiny architectures suffer from a large accuracy decrease. As Figure 1 shows, Evolved Transformer and Lite Transformer lose 2.9 and 2.4 BLEU score compared to base transformer, respectively.

## 3 DICTFORMER

**Overview.** We propose DictFormer with dictionary sharing to enable a fast, compact, and accurate transformer. When matching transformer's accuracy, DictFormer can reduce more than $3.6\times$ parameters and $\sim 3\times$ Mult-Adds shown in Figure 1, which outperforms prior transformers with a higher BLEU score. Also, DictFormer is flexible to compress model size given an accuracy threshold. For instance, when DictFormer matches the accuracy of Lite Transformer, it could further compress the model of lite transformer by $\sim 2.2\times$. Given a $N$-layer transformer model, we can easily transform it into DictFormer by converting all the weights in $N$ blocks into one shared dictionary and few unshared look-up coefficients, shown in Figure 2. For example, $N$-layer weights attention $W_i^A$ and FFN $W_i^F$, where $i \in [0, N-1]$, in Transformer (Figure 2(a)) are represented by smaller dictionaries $D^A$, $D^F$ that are shared by $N$ blocks, and $N$-layer unshared coefficients $C_i^A$ and $C_i^F$ for DictFormer (Figure 2(b)). Meanwhile, the attention and FFN operations in vanilla Transformer are replaced with shared-dictionary attention and group-wise shared-dictionary FFN whose details are introduced in our following contents. Dictformer reduces Transformer's #Params from $\mathcal{O}(d^2N)$ to $\mathcal{O}(d(m+tN))$. This is a $\mathcal{O}(dN/(m+tN))\times$ model size reduction since dictionary size $m < d$, coefficient size $t << d$, where the embedding size is $d$. Dictformer also reduces #Mult-Adds from $\mathcal{O}(d^2Nn)$ to $\mathcal{O}(dNn(m+t))$, where $n$ is input sequence length. The details on how to define and calculate #Params and #Mult-Adds are shown in Appendix A.1.

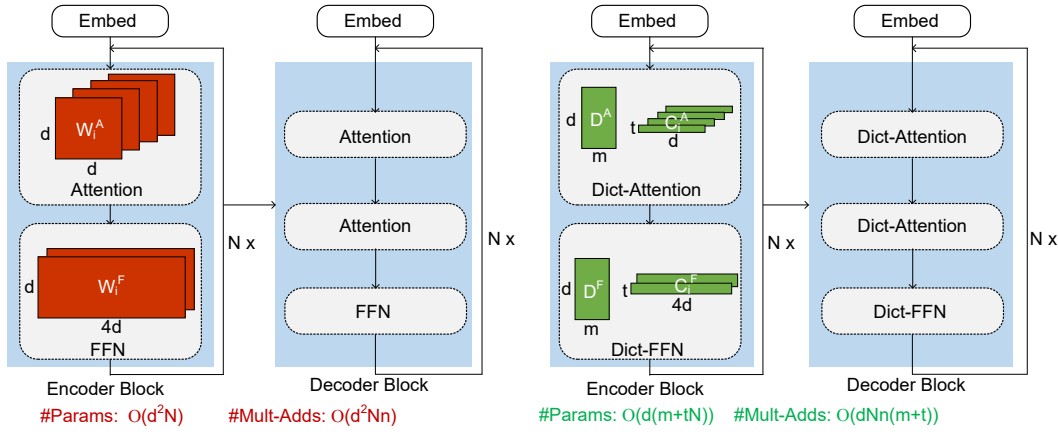

Figure 2: Our DictFormer replaces $N$-layer unshared, large attention and FFN weights $W_i^A$, $W_i^F$ ($i \in [0, N-1]$) in transformer with smaller, shared dictionaries $D^A$, $D^F$ and coefficients $C_i^A$, $C_i^F$. The Mult-Adds operations between weights and inputs in Transformer are also reduced by our dictionary look-ups and few linear projections with coefficients.

**Shared-dictionary Attention.** Given a $N$-layer transformer model, we define that $Q_i$, $K_i$, and $V_i$ are the $i$-th layer query, key, and values. Same to Transformer (Vaswani et al., 2017), we also use equation 1 to calculate the attention scores once we have query, key, and values.

$$Attention(Q_i, K_i, V_i) = softmax(\frac{Q_i \cdot K_i^T}{\sqrt{d}}) \cdot V_i \tag{1}$$

Our DictFormer utilizes equation 2 instead of $MultiHead(Q_i, K_i, V_i) = MH_i \cdot W_i^O$ used in Transformer (Vaswani et al., 2017) to compute multi-head values, where $MH_i$ is derived according to equation 3, and $D^A$, $C_i^O$, and $I_i^O$ are used to linearly project $MH_i$, instead of $W_i^O$. The attention value $head_i^j$ for each head $j$ in layer $i$ is computed by equation 4 instead of $head_i^j = Attention(Q_i \cdot W_i^{Q_j}, K_i \cdot W_i^{K_j}, V_i \cdot W_i^{V_j})$ used in existing transformers.

$$MultiHead(Q_i, K_i, V_i) = SD(MH_i, D^A, C_i^O, I_i^O) \tag{2}$$

$$MH_i = Concat(head_i^1, ..., head_i^h) \tag{3}$$

$$head_i^j = Attention(SD(Q_i, D^A, C_i^{Q_j}, I_i^{Q_j}), SD(K_i, D^A, C_i^{K_j}, I_i^{K_j}), SD(V_i, D^A, C_i^{V_j}, I_i^{V_j})) \tag{4}$$

The reason why we use the shared attention dictionary $D^A$, indices $I_i$ and coefficients $C_i$ to replace the larger and unshared weights $W_i^{Q_j}$, $W_i^{K_j}$, $W_i^{V_j}$ and $W_i^{O_j}$ used in previous transformers is that the linear projections of lookup of $D^A$ with coefficients $C_i^X$ can have similar representation abilities as $W_i^{X_j}$, where $X$ represents inputs type, e.g., $X = Q$ means input is query. To be specific, $D^A$, $I_i$, and $C_i$ have size of $d \times m^A$, $t^A \times d$, and $t^A \times d$, respectively. $m^A$ and $t^A$ are the dictionary size and coefficient size in attention. As equation 5 shows, the linear projections in transformer's attention, e.g., $W_i^{Q_j} \cdot Q_j$, are replaced by our lightweight shared dictionary projection function ($SD$), e.g., $SD(Q_j, D^A, C_i^{Q_j}, I_i^{Q_j})$, derived by equation 6 and equation 7. $SD(Q_j, D^A, C_i^{Q_j}, I_i^{Q_j})$ replaces $W_i^{Q_j} \cdot Q_j$ since $W_i^{Q_j}$ can be replaced by looking up $D^A$ and few linear projections with coefficients $C_i^{Q_j}$ without accuracy decrease. The following paragraphs and Figure 3(a) introduce the details of look-up and scaling using equation 6 and equation 7.

$$Q_j \cdot W_i^{Q_j} \Rightarrow SD(Q_j, D^A, C_i^{Q_j}, I_i^{Q_j}) \tag{5}$$

In equation 6, the lookup of $D^A$ by indices $I_i^{Q_j}$ is defined by $D^A[:, I_i^{Q_j}[t, i_d]]$ that can fetch the $I_i^{Q_j}[t, i_d]$-column vector from $D^A$; Unshared linear projection that is used to enlarge the representation ability of shared dictionary is depicted by $\widetilde{W_i^{Q_j}} = \sum_{t=1}^{t^A} C_i^{Q_j}[t, i_d] \odot D^A[:, I_i^{Q_j}[t, i_d]]$, where $\odot$ represents scaling a fetched vector from a dictionary with a scalar in coefficients. Therefore, linearly projecting $Q_j$ with $W_i$ according to $W_i^{Q_j} \cdot Q_j$ is replaced by $\widetilde{W_i^{Q_j}} \cdot Q_j$ in equation 6 when we find proper $D^A$, $I_i^{Q_j}$, $C_i^{Q_j}$ to meet that $\widetilde{W_i^{Q_j}}$ and $W_i^{Q_j}$ have similar representation abilities, e.g., they have matched accuracy. Directly generating $\widetilde{W_i^{Q_j}}$ and multiplying it with $Q_j$ potentially increases the computations. To tackle this problem, we compute the multiplications between $Q_i$ and dictionary $D^A$ as $O_i$ first according to equation 7, and reuse $O_i$ for the following look-ups and scaling with $C_i^{Q_j}$.

$$SD(Q_i, D^A, C_i^{Q_j}, I_i^{Q_j}) = Q_i \cdot \sum_{t=1}^{t^A} C_i^{Q_j}[t, i_d] \odot D^A[:, I_i^{Q_j}[t, i_d]]$$

$$= (\sum_{t=1}^{t^A} Q_i \cdot D^A[:, I_i^{Q_j}[t, i_d]]) \odot C_i^{Q_j}[t, i_d] \quad (6)$$

$$= \sum_{t=1}^{t^A} O_i[:, I_i^{Q_j}[t, i_d]] \odot C_i^{Q_j}[t, i_d], i_d \in [0, \frac{d}{h}]$$

$$O_i[:, b] = Q_i \cdot D^A[:, b], b \in [1, m^A] \quad (7)$$

We use Figure 3(a) to show an example of equation 5 about how to lookup dictionary with in-

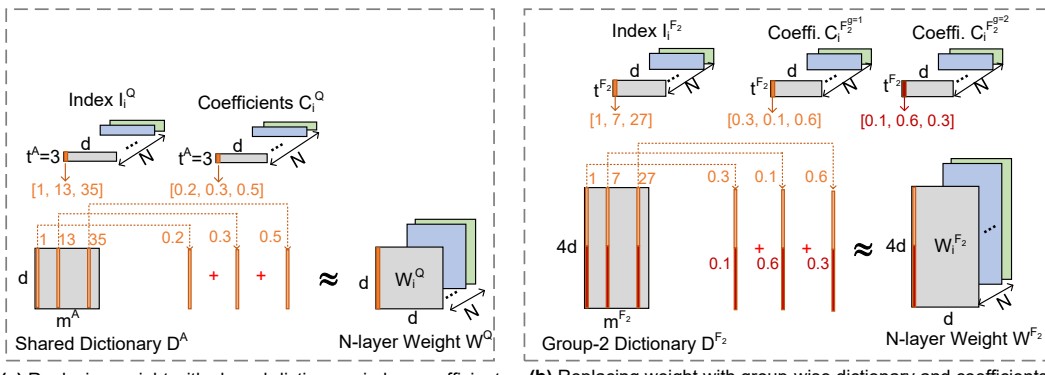

(a) Replacing weight with shared-dictionary, index, coefficients    (b) Replacing weight with group-wise dictionary and coefficients

Figure 3: (a) An example of looking up dictionary with indices and scaling it with coefficients. This lookup and scaling is able to reconstruct $N$-layer weight. (b) An example of looking up group-wise dictionary and its scaling for large-dimension representation projection.

dices and scaling it with coefficients. This process can create a $N$-layer weight $W^Q$. For example, given the shared dictionary $D^A$, to generate the first column of weight $W_i^Q$, the first column of index matrix $I_i^Q$ is taken out, e.g., $[1, 13, 35]$, and is used as indices to fetch the the corresponding columns from $D^A$, e.g., $D^A[:][1], D^A[:][13], D^A[:][35]$. Then the first column of coefficients $C_i^Q$, e.g., $[0.2, 0.3, 0.5]$, are multiplied with $D^A[:][1], D^A[:][13], D^A[:][35]$ respectively, and the sum of multiplication works as the first column of $W_i^Q$. In this way, weights $W$ in attention with $4d^2N$ parameters are compressed into $m^A$, $I^Q$, and $C^Q$ with size $dm^A + 8t^AdN$. For example, DictFormer can reduce $8.7\times$ model size when $d = 512$, $m^A = 256$, $t^A = 24$, $N = 6$. More calculation details can be found at appendix A.1.

**Group-wise Shared-dictionary FFN.** The FFN in prior transformer includes two-layer computations: (i) $F_1 = max(0, X_i \cdot W_i^{F_1} + b_1)$, (ii) $F_2 = F_1 \cdot W_i^{F_2} + b_2$. Instead of the regular linear projections $X_i \cdot W_i^{F_1} + b_1$ and $F_1 \cdot W_i^{F_2} + b_2$, DictFormer uses a new lightweight projection called group-wise shared dictionary projection (GSD), i.e., $GSD(X_i, D, C_i^{F_1}, I_i^{F_1}, G)$ and

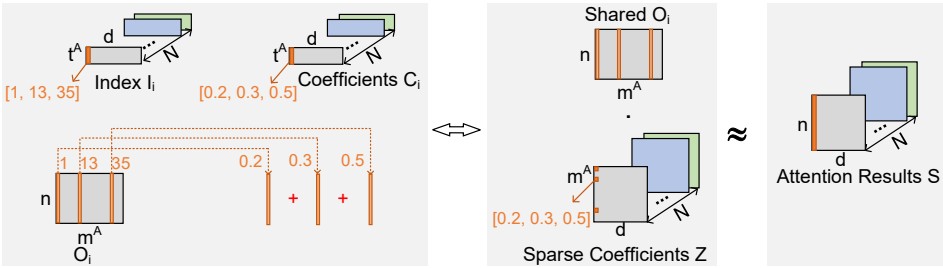

Figure 4: Since index $I$ in DictFormer is not differentiable, we train sparse coefficients Z instead of jointly training $I$ and $C$. After training, sparse $Z$ is converted to $I$ and $C$ for deployment.

$GSD(F_1, D, C_i^{F_2}, I_i^{F_2}, G)$, in equation 8 and equation 9 to compute the FFN. Here $GSD$ projection is same as $SD$ in shared-dictionary attention when $G = 1$. In $SD$ projection, it works well to replace a $N$ weights of $d \times d$ with a $d \times m$ dictionary where each column of dictionary is multiplied with a scalar in coefficients. However, when the column of dictionary is large, e.g., $4 \times d$ in the second layer of FFN, $SD$ projection is difficult to achieve an accurate model since an vector of dictionary with $4 \times d$, e.g. 2048, multiplied by a same scalar is not flexible enough. To increase the flexibility of $SD$ projection and improve performance, $GSD$ firstly divides each column of dictionary into $G$ groups equally and assign different scalars to multiply the numbers in each group.

$$F_1 = max(0, GSD(X_i, D, C_i^{F_1}, I_i^{F_1}, G) + b_1) \tag{8}$$

$$F_2 = GSD(F_1, D, C_i^{F_2}, I_i^{F_2}, G) + b_2 \tag{9}$$

Equation 10 and equation 11 show the computations of $GSD$. Dictionary $D^{F_2}[:, :]$ is divided equally into $G$ groups and the $g$-th group is defined as $D^{F_2}[Index_g, :]$, where $Index_g = [(g-1)\frac{4d}{G}, g\frac{4d}{G})$ and $g \in [1, G]$. Also, the multiplication result between $g$-th group dictionary and input is defined as $O^g[:, b]$ shown in equation 11, where $b \in [1, m^{F_2}]$. Different groups of dictionary share the same look-up indices $I_i^{F_2}$. The $t^{F_2}$ queried vectors in each group will be scaled by its corresponding coefficients $C_i^{F_2^g}$ and the scaled results are then accumulated to derive the $g$-th $GSD$ result shown in equation 10. In addition, $G$ group-wise $GSD$ are further summed to compute the final $GSD$ result.

$$GSD(X, D^{F_2}, C_i^{F_2}, I_i^{F_2}, G) = \sum_{g=1}^{G} \sum_{t=1}^{t^{F_2}} O^g[:, I_i^{F_2}[t, i_d]] \odot C_i^{F_2^g}[t, i_d] \tag{10}$$

$$O^g[:, b] = X[:, Index_g] \cdot D^{F_2}[Index_g, b], g \in [1, G], b \in [1, m^{F_2}], Index_g = [(g-1)\frac{4d}{G}, g\frac{4d}{G}) \tag{11}$$

Figure 3(b) shows an example of group-2 shared dictionary for the the second layer of FFN. The dictionary $D^{F_2}$ is equally split into two parts that share the same indices $I_i^{F_2}$ but each group has unique coefficients, e.g., $C_i^{F_2^{g=1}}$ and $C_i^{F_2^{g=2}}$. The first group and the second group use the corresponding coefficients, e.g., $[0.3, 0.1, 0.6]$ and $[0.1, 0.6, 0.3]$, to scale the queried dictionary vectors, respectively. The accumulation of scaled vectors can be used to represent the $N$-layer weight $W^{F_2}$.

**Training DictFormer via Constraints and Relaxation.** DictFormer replaces transformer weights with linear projections of a dictionary, thereby having three-step computations including small projections, lookup and scale. DictFormer computes a small multiplication between input and dictionary, generating intermediate variable $O$, looking up $O$ and scaling it with coefficients. To train the DictFormer, one should jointly optimize dictionary, index $I$ and coefficients $C$. Directly training the Dictformer leads to a combinatorial optimization problem since index $I$ is non-continuous. Although one could use AutoML, such as evolutionary method and Reinforcement learning to jointly learn dictionary, $I$ and $C$, these methods suffer from a large training time with low performance. To workaround the training of index $I$, we reformulate the method to a regular linear projection with sparse constraints, which can efficiently train the DictFormer.

We convert the index $I_i$ and coefficients $C_i$ into sparse coefficients $Z$, so that the training of index $I_i$ and coefficients $C_i$ is replaced with training sparse coefficients $Z$. During the deployment phase, the sparse coefficients $Z$ are reversely transformed into index $I_i$ and coefficients $C_i$ shown in Figure 4. In particular, the shape of coefficients $C$ and index $I$ is $t^A \times d \times N$. The shape of sparse $Z$ is $m^A \times d \times N$. There are two steps to derive $Z$: Initializing all elements as 0 and copying $C$ elements to $Z$ according to the index values in $I_i$. For example, since the first column of $I_1$ and $C_1$ is $[1, 13, 35]$ and $[0.2, 0.3, 0.5]$, all entries of the first column in $Z$ are zeros except the 1st entry, 13th entry, and the 35th entry are $0.2$, $0.3$, and $0.5$. Therefore, the lookup and scale of $O_i$ can be converted into the matrix multiplication between $O_i$ and $Z_i$ shown in equation 12. The new coefficient $C$ has a sparsity constraint that requires that the non-zero elements ($l_0$ $norm$) of each column is $t_A << m^A$ shown in equation 13. Now we can train $Z$ with $||Z_i^{Q_j}[:, i_d]||_{l_0} = t^A$, instead of training $I$ and $C$.

$$\sum_{t=1}^{t^A} O_i[:, I[t, i_d]] \odot C_i^{Q_j}[t, i_d] = O_i \cdot Z_i^{Q_j} \tag{12}$$

$$||Z_i^{Q_j}[:, i_d]||_{l_0} = t^A \tag{13}$$

The $l_0$ $norm$ sparsity constraint in equation 13 is non-differentiable and we cannot directly train $Z$. To achieve the training with sparsity, we are inspired by existing sparse training methods Tsuruoka et al. (2009); Li et al. (2016) to firstly loose this constraint to a $l_1$ $norm$ constraint shown in equation 14 to penalize the non-zero parameters and clip the unimportant values that are near to zeros using equation 16. To be specific, the gradients of coefficients are calculated by equation 15, where parameter $\lambda$ is used to control the trade-off between the gradient of loss $L$, $\frac{\delta L}{\delta Z}$, and $l_1$ $norm$ sparsity constraint.

$$||Z||_{l_1} = \sum_{i_d}^{d} ||Z_i^{Q_j}[:, i_d]||_{l_1} \tag{14}$$

$$\frac{\delta(L + \lambda||Z||_{l_1})}{\delta Z} = \frac{\delta L}{\delta Z} + \lambda sign(Z) \tag{15}$$

Equation 16 is a threshold function during training. During the backpropagation, the derivative of threshold function is 1 if $|x| > value(\rho)$, 0 otherwise. During the forward phase, Equation 16 can be used to globally clip the near-zero values to zero given a ratio $\rho$, where $value(\rho)$ derives the value at ratio $\rho$ in the ascending order. One also can force that at most $t$ of the elements in each column of $Z$ are non-zero while the rest are set to 0 by letting $value(\rho)$ is the $t$-th largest value of each column.

$$\mu(x) = \begin{cases} x, & \text{if } |x| > value(\rho). \\ 0 & \text{otherwise.} \end{cases} \tag{16}$$

## 4 EXPERIMENTAL METHODOLOGY

**Machine Translation Dataset.** Three machine translation benchmarks are tested: IWSLT'14 German-English (De-En), WMT'14 English to German (En-De), and WMT'14 English to France (En-Fr). For IWSLT'14 De-En, we adopt the same settings in Wu et al. (2020) with 160K training sentence pairs and 10K joint byte pair encoding (BPE) vocabulary in lower case. For WMT'14 En-De, our models are trained with 4.5M sentence pairs, validated on newstest2013, and tested on newstest2014. Also, a 32K joint source and target BPE is used in the vocabulary. For WMT'14 En-Fr, we follow the setup in Wu et al. (2020) by training models on 36M training sentence pairs from WMT'14, validating on newstest2012 and 2013, and testing on newstest2014. The vocabulary has a size of 40K and is based on a joint source and target BPE factorization. The evaluation setting used in Vaswani et al. (2017) is utilized with a beam size of 4 and a length penalty of 0.6. We replicate the same BLEU calculations in Wu et al. (2020) with case-sensitive tokenization. The last 10 model checkpoints are averaged for testing and the lowest-perplexity model is picked for validation.

**Abstractive Summarization Dataset**. We evaluate DictFormer on CNN-DailyMail dataset (Chen et al., 2016) that has 280K news articles with multi-sentence summaries. We follow the settings

in (Wu et al., 2020), truncate the articles to 3000 tokens, use a 30K BPE vocabulary and F1-Rouge as the metric.

**Language Modeling Dataset**. We also evaluate DictFormer on WIKITEXT-103 (Merity et al., 2016) that has a 260K BPE vocabulary and contains 103M/217K/245K tokens for training, validation, and testing. Language modeling performance about perplexity (ppl) is reported.

**Architecture and Evaluation.** For machine translation tasks, we follow the sequence-to-sequence transformer (Vaswani et al., 2017) with encoder and decoder to develop our model. For IWSLT dataset, the general settings are same as (So et al., 2019). For WMT dataset, the setting is based on (Wu et al., 2020). Also, the same model for machine translation is used for summarization task. For language modelling, we use the transformer-based decoder architecture of (Baevski & Auli, 2019) but with 512 model dimensions and 12 blocks that is same to (Wu et al., 2020). Fairseq's transformer implementation (Ott et al., 2019) is used as the backbone for the baseline model.

In our DictFormer architecture, encoder and decoder have three dictionaries, respectively, *i.e.*, encoder attention dictionary $D_{enc}^A$, encoder $F_1$ dictionary $D_{enc}^{F_1}$, encoder $F_2$ dictionary $D_{enc}^{F_2}$, and decoder attention dictionary $D_{dec}^A$, decoder $F_1$ dictionary $D_{dec}^{F_1}$, decoder $F_2$ dictionary $D_{dec}^{F_2}$. Each of them is shared by all blocks in encoder or decoder. For example, $D_{enc}^A$ is shared by all attention blocks in encoder. We study the effects of DictFormer with various dictionary size from 80 to 240 and different coefficient size from 12 to 84 shown in Figure 8. One can pick up a dictionary size and coefficient size to control the trade-off between accuracy and model size according to the requirements. For example, one could get the results in our tables when dictionary size is set as $m^A = \frac{d}{2.5}$, $m^{F_1} = \frac{d}{2.5}$, $m^{F_2} = \frac{d}{2}$, and $t^A = \frac{d}{10}$, $t^{F_1} = t^{F_2} = \frac{d}{15}$.

In this paper, #Params means parameter numbers of transformer without including embedding layers; #TParams shows parameter numbers of model and embedding layers. The Mult-Adds calculation is tested when a model is used to translating two sentences with the length of 30 in default (same to previous work Wu et al. (2020)).

Table 1: Comparison with state-of-the-art transformers on machine translation tasks. For a fair comparison with existing tiny transformer, We follow Lite Transformer (Wu et al., 2020) to scaling down the hidden size to meet the deployment requirements on mobile settings. For example, Transformer uses a 128 hidden size and 2.8M #Params; #Ops represents Mult-Adds and Entries with $\pm$ shows the average across three independent runs.

| | IWSLT'14 De-En | | | | | WMT'14 En-De | | | | | WMT'14 En-Fr | |
| --- | --- | --- | --- | --- | --- | --- | --- | --- | --- | --- | --- | --- |
| | #Params | Ratios | BLEU | $\triangle$BLEU | #Ops | #Params | Ratios | BLEU | $\triangle$BLEU | #Ops | BLEU | $\triangle$BLEU |
| Transformer (Vaswani et al., 2017) | 2.8M | 1.0× | 27.8 | - | 63M | 2.8M | 1.0× | 21.3 | - | 87M | 33.6 | - |
| Universal Trans (Dehghani et al., 2019) | 2.1 M | 1.5× | 30.3 | +2.5 | 152M | 2.6M | 1.1 × | 22.4 | +1.1 | 193M | 35.8 | +2.2 |
| Tensorized. Trans (Ma et al., 2019) | 1.4 M | 2× | 28.0 | 0.2 | - | 1.7M | 1.6× | 21.2 | -0.1 | - | 33.4 | -0.2 |
| Lite Transformer (Wu et al., 2020) | 2.3 M | 1.2× | 29.6±0.4 | +1.8 | 46M | 1.8 M | 1.6× | 21.8±0.3 | +0.6 | 47M | 35.3±0.4 | +1.7 |
| DeLighT (Mehta et al., 2021) | 1.1 M | 2.5× | 28.9±0.2 | +1.1 | 49M | 1.6M | 1.8× | 21.8±0.1 | +0.5 | 64 M | 34.2±0.2 | +0.6 |
| Subformer (Reid et al., 2021) | 1.3M | 2.2× | 27.6±0.3 | -0.2 | 61M | 1.9M | 1.5 × | 21.0±0.2 | -0.3 | 85M | 33.5±0.4 | -0.1 |
| **DictFormer** | **0.3 M** | **8.9×** | **30.0**±0.3 | **+2.2** | **32M** | **0.6 M** | **4.9×** | **22.0**±0.2 | **+0.7** | **46M** | **35.6**±0.3 | **+2.0** |

## 5 RESULTS

**Machine Translation.** We first report the machine translation results and compare them with prior works. Our baseline transformer model settings are in line with Lite Trans. (Wu et al., 2020) which provides the best results under mobile settings. Our DictFormer generally outperforms the state-of-the-art transformers on IWSLT'14 De-En, WMT'14 En-De, and WMT'14 En-Fr. Table 1 represents the quantitative results of our DictFormer. Compared to Transformer (Vaswani et al., 2017), Dict-Former compresses $8.9\times$ model size, reduces $1.9\times$ #Mult-Adds, and improves 2.2 BLEU score on IWSLT'14 De-En, which achieves better trade-off between performance and model size than previous works. Similar to IWSLT'14 De-En, our experiments on larger corpora including WMT'14 En-De and En-Fr also show DictFormer obtains more compact and accurate model than existing Transformers. For instance, only our DictFormer can compress more than $6\times$ model size over Transformer without decreasing any model performance under mobile settings.

Table 2: Comparison with state-of-the-art transformers on abstractive summarization and Language modeling tasks. #Ops is calculated by longer sequence with the input length of 100. Entries with $\pm$ represents the average across three independent runs.

| | CNN-DailyMail for abstractive summarization | | | | | | WIKITEXT103 for language modeling | | | | |
| | R1 | R2 | RL | #Ops | #Params | Ratios | Valid ppl. | Test ppl. | #Ops | #Params | Ratios |
|---|---|---|---|---|---|---|---|---|---|---|---|
| Trans./Adaptive | 41.4 | 18.9 | 38.3 | 3.6G | 41.4M | 1.0 $\times$ | 23.2 | 24.0 | 50.3G | 37.8M | 1.0$\times$ |
| Lite Transformer | 41.3 | 18.8 | 38.3 | 1.5G | 17.3M | 2.4 $\times$ | 21.4 | 22.2 | 48.7G | 37.2M | 1.01$\times$ |
| DelighT | - | - | - | - | - | - | 23.5 | 24.1 | - | $\sim$33M | 1.1$\times$ |
| Subformer | 41.6 | 19.2 | 38.4 | - | 41M | 1.01 $\times$ | 20.8 | 21.1 | $\sim$50G | $\sim$18M | 2.1$\times$ |
| **DictFormer** | **41.3**$\pm$0.2 | **18.9**$\pm$0.1 | **38.3**$\pm$0.1 | **0.9G** | **8.9M** | **4.7**$\times$ | **21.3**$\pm$0.2 | **22.2**$\pm$0.2 | **20.3G** | **10.0M** | **3.8**$\times$ |

**Abstractive Summarization and Language Modeling.** We then report the abstractive summarization and language modeling results and compare them with prior works. Abstractive summarization model follows the transformer model, but language modeling model follows the adaptive inputs model (Baevski & Auli, 2019). Our DictFormer consistently outperforms prior transformers on both summarization and language modeling tasks. Table 2 represents the quantitative results of our DictFormer. For abstractive summarization on CNN-DailyMail, DictFormer achieves similar F1-Rouge score with Transformer but requires $4.7\times$ less model size and $\sim 4\times$ less computations. For language modeling on WIKITEXT103, compared to adaptive inputs model (Baevski & Auli, 2019), DictFormer compresses $3.8\times$ model size with less computations and matched testing perplexity.

**Sensitive Study and Ablation Study.** We study the DictFormer's performance under different model size and hyper-parameters in Appendix. We also use table 3 in Appendix to show the ablation study of DictFormer's techniques. The performance of DictFormer improves with an increase in the number of model parameters, across different corpora. DictFormer achieves similar BLEU score with Transformer using fewer model parameters. Although DictFormer's performance improves with an increase in the size of dictionary and coefficients, the model size is also enlarged. So one should pick up a proper dictionary or coefficient size according to accuracy requirements.

# 6 CONCLUSION

We propose DictFormer with dictionary sharing to enable a fast, compact, and accurate transformer. DictFormer significantly reduces both model size and computation parameters of prior transformers with similar performance. Also, DictFormer can consistently improve the performance on multiple tasks, such as machine translation, abstractive summarization, and language modeling with similar model size. DictFormer code is available at `https://github.com/SamNLP/DictFormer`.

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

# A APPENDIX

## A.1 DICTFORMER BENEFITS WITH DICTIONARY SHARING

**The block comparison of Transformer and our DictFormer.**

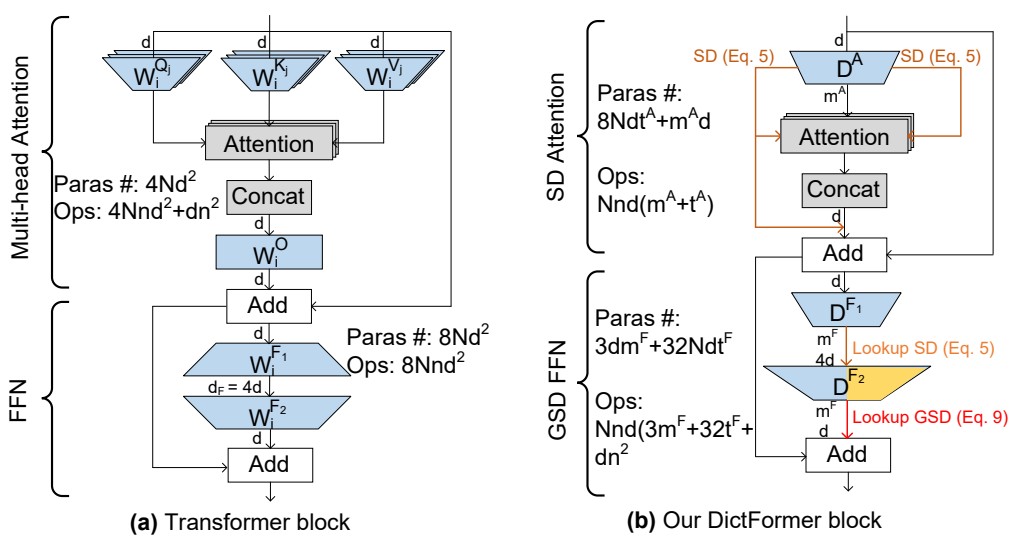

Figure 5: The block comparison of (a) transformer and (b) our DictFormer.

Figure 5 further depicts the comparison of transformer encoder block and DictFormer encoder block. The $i$-th Transformer block including attention and FFN has four weights $W_i^A$ and $W_i^F$. $W_i^A$ has four weights including query $W_i^{Q_j}$, key $W_i^{K_j}$, value $W_i^{V_j}$, and output $W_i^{O_j}$, thereby having $4Nd^2$ parameters and $4Nnd^2$ Mult-Adds given a sequence of size $n$. $W_i^F$ has two parts: $W_i^{F_1}$ and $W_i^{F_2}$. $N$-layer FFN blocks have $8Nd^2$ parameters and $8Nnd^2$ Mult-Adds operations. In contrast, Dict-Former significantly reduces the parameters and Mult-Adds of Transformer. Specifically, each Dict-Former block replaces the attention and FFN with shared dictionary (SD) attention and group-wise shared dictionary (GSD) FFN. Linear projection is performed by looking up dictionary $D^A$ using Eq.5. SD Attention has $8Ndt^A + m^A d$ parameters and $Nnd(m^A + t^A)$ operations. GSD FFN has $3dm^F + 32Ndt^F$ parameters and $Nnd(3m^F + 32t^F)$ operations, where $t$, $m$ are the size of coefficient and dictionary. To simplify the calculations of #params and #Mult-Adds, we can let $t^A = t^F$, $m^A = m^F$. Dictformer reduces Transformer's #Params from $\mathcal{O}(d^2N)$ to $\mathcal{O}(d(m + tN))$. This is a $\mathcal{O}(dN/(m + tN))\times$ model size reduction since dictionary size $m < d$, coefficient size $t << d$, where the embedding size is $d$. Dictformer also reduces #Mult-Adds from $\mathcal{O}(d^2Nn)$ to $\mathcal{O}(dNn(m + t))$.

## A.2 THE COMPARISON BETWEEN WEIGHT SHARING AND DICTIONARY SHARING

Weight sharing is surprisingly effective to compress model size for discriminate NLP models based on Transformer encoders, e.g., BERT. For example, prior work ALBERT (Lan et al., 2020) shows that even sharing all parameters across layers does not introduce any accuracy reduction. However, for generative sequence-to-sequence models based on transformer's encoders and decoders, sharing all parameters will significantly decrease accuracy on multiple standard machine translation or language modelling tasks (Reid et al., 2021; Takase & Kiyono, 2021). To match vanilla Transformer's accuracy, multiple works (Reid et al., 2021; Takase & Kiyono, 2021; Xia et al., 2019; Ma et al., 2019) only share weights across partial layers instead of all layers. However, partial weights sharing remarkably brings down the model size compression effect of weights sharing. Also, how to decide which layers should be shared in partial weights sharing is difficult due to the large and dynamic search space that is dependent on the specific tasks.

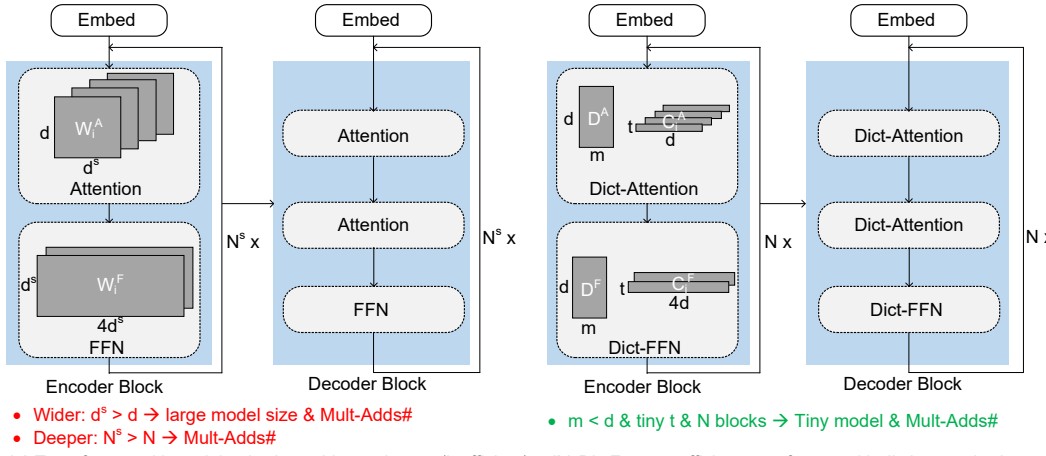

- Wider: $d^s > d$ → large model size & Mult-Adds#
- Deeper: $N^s > N$ → Mult-Adds#

- m < d & tiny t & N blocks → Tiny model & Mult-Adds#

**(a)** Transformer with weight sharing: wider or deeper (inefficient)    **(b)** DictFormer: efficient transformer with dictionary sharing

Figure 6: (a) Transformer with weights sharing. It contains three parts including embedding, $N^s$ encoder blocks and $N^s$ decoder blocks. Each encoder/decoder block contains attention and Feed-Forward Network (FFN). The embedding size is $d^s$ and FFN feature size is $4 \times d^s$. Weights in the $i$-th Attention and FFN are denoted as $W_i^A$ and $W_i^F$. To match or improve accuracy of transformer w/o weights sharing, transformer w/ wights sharing should be wider ($d^s > d$) or deeper ($N^s > N$), where $d$ and $N$ are the embedding size and blocks number of transformer w/o wight sharing. (b) Our DictFormer with dictionary sharing. $W_i^A$ and $W_i^F$ are represented by smaller dictionaries $D^A$, $D^F$, and coefficients $C_i^A$ and $C_i^F$, where dictionary size $m < d$, coefficient size $t << d$.

Transformer with all-parameters sharing such as Universal Transformer (Dehghani et al., 2019) matches or improves transformer's performance at the cost of a wider or deeper transformer architecture. A wider transformer with a larger embedding dimension enlarges the model size and brings larger computations (Mult-Adds). A deeper transformer with more encoder/decoder blocks does not only increase model size, but also introduces more computations. Importantly, weights sharing techniques cannot reduce Mult-Adds numbers and training/inference time. Figure 6(a) shows the comparisons of Transformers with weights sharing and our DictFormer with dictionary sharing. Weights sharing techniques cannot solve the deployment challenges of transformers on resource-limited devices and obtain real-time NLP applications. To reduce both model size and computations of Transformers, we introduce DictFormer with dictionary sharing instead of previous weights sharing. In particular, DictFormer shares dictionary across all layers so there is no need to decide which layers should be shared. Also, DictFormer with few unshared look-up coefficients does not require a wider embedding size or deeper encoder/decoder to improve accuracy. Rather than compressing model size by parameters sharing, DictFormer with dictionary sharing can also enable computations sharing to reduce running latency. Therefore, our DictFormer provides a compact, fast, and accurate transformer model for sequence-to-sequence NLP tasks.

## A.3 TRAINING SETTINGS

To fairly compare with our baselines, all of our training settings are in line with (Wu et al., 2020). For machine translation tasks, a dropout of 0.3 is used and the dropout ratio is linearly scaled down when we shrink the dimension of the embeddings for WMT datasets. The learning rate linearly warms up from $10^{-7}$ to $10^{-3}$ followed by a cosine annealing with a single cycle and Adam optimizer same as (So et al., 2019) and (Wu et al., 2020). Inverse linear square root learning rate scheduling is used for IWSLT De-En. The summarization training settings are the same as machine translation. To train models on language modeling task, we follow the training setting in (Baevski & Auli, 2019), but decrease the dropout rate by half in FFN layers. The training experiments of WMT, summarization, and language modeling are conducted on 8 NVIDIA Tesla V100 GPUs. IWSLT De-En is trained on two GPUs. Machine translations tasks are trained for 50K steps, but language modelling tasks are trained for 286K steps. We further describe deployment settings in Appendix.

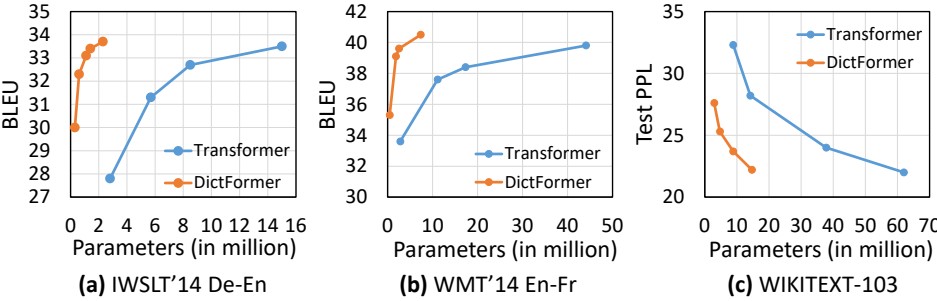

Figure 7: The performance of DictFormer improves with an increase in the number of model parameters, across different corpora.

## A.4 DEPLOYMENT SETTINGS.

Deploying deep learning models including transformer on powerful cloud-based servers as a service suffers from data privacy issues when users' data is private or confidential (Lou & Jiang, 2019; Lou et al., 2019). In addition, the inference cost of privacy-sensitive applications that performed over encrypted data using fully homomorphic encryption is prohibitively expensive (Feng et al., 2021; Lou et al., 2021; 2020a). (i) Mobile settings. Scaling up transformer models leads to higher machine translation performance, but those large architectures are not suitable for real-world applications, especially when we should deploy them on edge devices where computation and memory size are highly constrained. Multiple prior works like (Wu et al., 2020) formalize the deployment of lightweight transformers on mobile devices by defining the mobile settings regarding the amount of computation and parameter numbers. Particularly, given the ARM Cortex-A72 mobile CPU with computation performance of 48G FLOPS, the computation constraint for machine translation should be under 500M Mult-Adds and the parameter number should be around 10M with a sequence of 30 tokens that are the general length for machine translation. (ii) System-independent settings. Other than deployed on mobile devices, DictFormer also supports system-independent deployment. In this setting, we directly choose prior transformer architectures like DelighT (Mehta et al., 2021) and Subformer (Reid et al., 2021) as baselines and compare our DictFormer with them under the the same performance constraint.

## A.5 SENSITIVITY STUDY

. We study the DictFormer's performance under different model size as Figure 7 shows. This is important since system-independent settings require scalable DictFormer. The performance of Dict-Former improves with an increase in the number of model parameters, across different corpora. DictFormer achieves similar BLEU score with Transformer using $3\times$ to $10\times$ less model parameters. In Figure 8, we also report the hyper-parameter's effects on DictFormer's performance with WMT'14 En-Fr dataset with embedding size $d = 512$. Figure 8(a) shows the dictionary size study when the coefficient size is fixed to 60. All dictionaries share the same size, and all coefficients also share same size. The Figure 8(b) shows the coefficient size study when the dictionary size is fixed 240. Although DictFormer's performance improves with an increase in the size of dictionary and coefficients, the model size is also enlarged. So one should pick up a proper dictionary or coefficient size according to accuracy requirements.

## A.6 ABLATION STUDY

We also use table 3 to show the ablation study of DictFormer's techniques. **Transformer** Vaswani et al. (2017) adopts the base architecture with hidden size 512. **Transformer-Flatten** flattens the feed-forward hidden size of transformer from 2048 to 512, thus reducing the parameters from 44 millions to 25 millions without hurting the accuracy. **Only-Dictionary** is based on Transformer-Flatten and uses unshared dictionary architecture. This method achieves 33.2 BLEU with 9.7 million model parameters. The #TParams is 14.3 million and 27.4 million when adding embedding parameters on IWSLT'14 De-En and WMT'14 En-Fr, respectively. When sharing dictionaries in attention, **Shared-Attention** achieves similar BLEU score with 4.2 million model parameters. Mean-

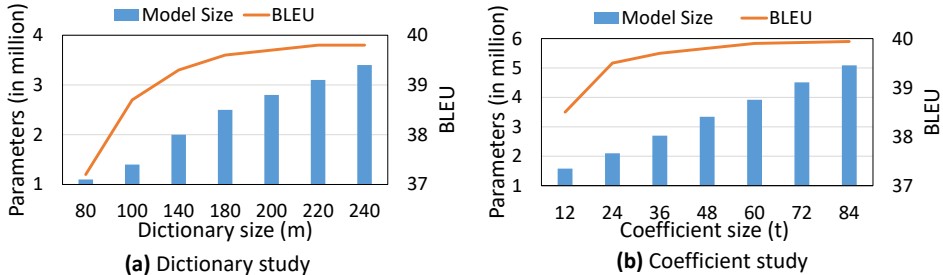

Figure 8: Increasing the size of dictionary (a) and coefficients (b) improves DictFormer's performance. (a) and (b) are tested on WMT'14 En-Fr dataset for machine translation task.

while, **Shared-FFN** only shares FFN using group-wise shared dictionaries (G is group number). DictFormer that shares both FFN and attention with G=2, denoted by **Shared-both**, requires 2.6M model parameters and matches the accuracy of unshared DictFormer. **Improved-Embed** represents that the embedding layers of Shared-both are further compressed by using existing method Mehta et al. (2021) so that the #TParams is reduced to 5.1M from 7.1M.

Table 3: Ablation study of DictFormer techniques on IWLS'14 De-En and WMT'14 En-Fr. Entries with $\pm$ represents the average across three independent runs.

| | IWSLT'14 De-En | | | WMT'14 En-Fr | | |
|---|---|---|---|---|---|---|
| | #Params | #TParams | BLEU | #Params | #TParams | BLEU |
| Transformer (Vaswani et al., 2017) | 44M | 48.6M | 34.4 | 44M | 64M | 40.7 |
| Transformer-Flatten | 25M | 29.6M | 34.4 | 25M | 45M | 40.7 |
| Only-Dictionary | 9.7M | 14.3M | 33.5 $\pm$0.3 | 10.6M | 27.4M | 39.5 $\pm$0.3 |
| Shared-Attention | 4.2M | 8.7M | 33.3 $\pm$0.4 | 4.5M | 21.3M | 39.5 $\pm$0.4 |
| Shared-FFN (G=1) | 7.8M | 12.2M | 33.2 $\pm$0.5 | 8.6M | 25.4M | 39.2 $\pm$0.5 |
| Shared-FFN (G=2) | 8.1M | 12.5M | 33.2 $\pm$0.3 | 8.9M | 25.7M | 39.6 $\pm$0.4 |
| Shared-FFN (G=3) | 8.4M | 12.8M | 33.3 $\pm$0.3 | 9.1M | 25.9M | 39.5 $\pm$0.3 |
| Shared-both (G=2) | 2.6M | 7.1M | 33.2 $\pm$0.4 | 2.5M | 19.3M | 39.6 $\pm$0.4 |
| Improved-Embed | 2.6M | 5.1M | 33.1 $\pm$0.4 | 2.5M | 9.8M | 39.5 $\pm$0.3 |

