# OpenReview forum: "DictFormer: Tiny Transformer with Shared Dictionary"
_ICLR.cc/2022/Conference — ICLR 2022 Poster_

### Official Review · Reviewer_uPqt · 2021-10-28

**Correctness:** 4
**Technical Novelty And Significance:** 3
**Empirical Novelty And Significance:** 3
**Recommendation:** 6
**Confidence:** 3

**Details Of Ethics Concerns:**

I have no concerns.

**Main Review:**

Strengths
- Clear description of background knowledge.
- Clear exposition of the proposed model.
- The authors perform a comprehensive comparison on different downstream tasks, such as, machine translation, summarization, and language modeling.
- The findings show that the proposed transformer model outperforms related work on the machine translation and language modeling tasks.

Weaknesses
- It is not clear how the initialisation of hyper-parameters affects model performance.

Questions to the Authors.
Please address the following questions during the rebuttal:

- Does parameter initialization could affect model performance? A possible extra contribution is to perform multiple  random runs and report variance. However, how expensive could this exercise become?
- Please speculate on how attention representations behave across layers. For example, in Abnar, and Zuidema Quantifying Attention Flow in Transformers or Voita, et al. The Bottom-up Evolution of Representations in the Transformer: A Study with Machine Translation and Language Modeling Objectives
- By using other pre-training objectives in the langgue modelling task (e.g. next sentence) would it change any finding or results?


**Summary Of The Paper:**

The authors proposed an efficient transformer layer based on a dictionary of shared parameters instead of standard self-attention.  The goal is to reduce redundant parameters in transformer models.  The main contributions are: a lite transformer model, modification of the self-attention parameters, and evaluation on language dowstream tasks. The proposed transformer model outperforms related work on the machine translation and language modelling tasks.

**Summary Of The Review:**

I recommend acceptance given that the paper clearly describes related work, and proposed model. The authors proposed an efficient transformer model that can be trained with less resources. The authors perform an  evaluation of the proposed model with different language downstream tasks, and the model outperforms related work on machine translation and language modelling.

---

> ### Author Response · Authors · 2021-11-19
> **Reply to Reviewer uPqt**
>
> We would like to thank again reviewer uPqt for the thoughtful comments and efforts towards improving our manuscript. Hope our following replies help answer the reviewer's outstanding questions.
>
> ### Question 1: It is not clear how the initialization of hyper-parameters affects model performance. A possible extra contribution is to perform multiple random runs and report variance.
>
> All coefficients in our experiments are initialized randomly using the same method as our baselines for a fair comparison.
>
> ### Question 2: Please speculate on how attention representations behave across layers. For example, in Abnar, and Zuidema Quantifying Attention Flow in Transformers or Voita, et al. The Bottom-up Evolution of Representations in the Transformer: A Study with Machine Translation and Language Modeling Objectives.
>
> Our empirical results show that DictFormer reduces $4.9\times$ to $8.9\times$ model size with similar accuracy over multiple tasks, compared to Transformer. We hypothesis that DictFormer should have similar attention representations as our baseline transformer since they have similar accuracy and they use the same attention scheme ( w.r.t DictFormer uses compact linear projections). We believe the approach in *Abnar, and Zuidema Quantifying Attention Flow in Transformers or Voita, et al }* will help verify our hypothesis by speculating and comparing the attention representations in DictFormer and Transformer.  This should be one of our future works.
>
> ### Question 3: By using other pre-training objectives in the language modeling task (e.g. next sentence) would it change any findings or results?
>
> We have high confidence that our findings are still similar if we use other pre-training objectives in a new language modeling task. This is because our experiments have shown consistent results on multiple tasks, including machine translation tasks, language modeling tasks, and abstractive summarization tasks.

---

> > ### Author Response · Authors · 2021-11-22
> > **Available for Discussion**
> >
> > We hope our responses have addressed the reviewer's concerns, but if not we are available/open to address any outstanding issues.

---

> > ### Comment · Reviewer_uPqt · 2021-11-26
> > **Questions addressed**
> >
> > Thank you for addressing my comments and questions. I have no further questions.

---

> > > ### Author Response · Authors · 2021-11-28
> > > **Thanks for your reviews and discussions**
> > >
> > > Dear reviewer uPqt,
> > >
> > > Happy to know your questions are addressed! Thanks again for your helpful reviews and suggestions!
> > >
> > > Bests,
> > >
> > > Paper2101 Authors

---

### Official Review · Reviewer_jV7A · 2021-11-02

**Correctness:** 3
**Technical Novelty And Significance:** 4
**Empirical Novelty And Significance:** 4
**Recommendation:** 6
**Confidence:** 4

**Main Review:**

### Strengths
- The proposed modification to the Transformer architecture reduces the number of model parameters and computational operations while sustaining competitive performance on various downstream tasks.
- To the best of my knowledge, the idea of replacing layers of the Transformer with shared dictionaries is novel.

### Room for Improvement

*Shared-dictionary Attention*
- I might be missing something but why is it stated that the unshared linear projection $\tilde{W_{i}^{Q_{j}}}$ is approximately equal to $W_{i}^{Q_{j}}$? My understanding is that this is not directly optimized for in the model.

*Group-wise Shared Dictionary FFN*
- The motivation behind dividing columns of the dictionary into groups is a bit unclear. What is meant by “high-quality performance” of the shared dictionary projection? Also, have the authors considered using a larger number of dictionary elements $m$ to increase the “flexibility” of the model?
- How is the number of groups $G$ determined?

*Training the DictFormer*
- Since the sparse matrix Z is initialized using values in $C$, how are coefficients $C$ initialized?

*Results tables*
- Missing confidence intervals. Were the experiments run with multiple seeds?

*Suggested related work*
- How is this work related to work on Sparse Transformers (e.g. [1], [2]) or fixed attention such as [3], [4]?

[1] Child R, Gray S, Radford A, Sutskever I. Generating long sequences with sparse transformers. arXiv preprint arXiv:1904.10509. 2019 Apr 23.

[2] Correia, G.M., Niculae, V. and Martins, A.F., 2019. Adaptively sparse transformers. arXiv preprint arXiv:1909.00015.

[3] You, W., Sun, S. and Iyyer, M., 2020. Hard-coded gaussian attention for neural machine translation. arXiv preprint arXiv:2005.00742.

[4] Raganato, A., Scherrer, Y. and Tiedemann, J., 2020. Fixed encoder self-attention patterns in transformer-based machine translation. arXiv preprint arXiv:2002.10260.

*Additional questions*
- Is it necessary to have the dictionary size less than the embedding size, namely $m < d$? Would it not be feasible to have a large dictionary ($m > d$) but keep the number of selected components $t$ small (i.e. $t < d$) through a sparsity constraint?
- Have the authors tracked whether all columns of the dictionaries are used in practice?
- Have the authors tracked what percentage of the $t$ coefficients are non-zero on average?

*Nitpicks*

Typos:
- p. 2, first line: “few unshared linear projection*s*”
- p. 3, “Overview” paragraph: “given a*n* accuracy threshold”
- p. 4, paragraph starting with “The reason why...”: “C_{i}^{x}” - should not $x$ be capitalized?
- p. 5, “Group-wise Shared-dictionary FFN” paragraph: “a $N d \times d$ weights” -> “$N$ weights of size $d \times d$”
- p. 6, Figure 4: “training sparse coefficients” -> “we train sparse coefficients”
- p. 6, first sentence of “Training DictFormer via Constraints and Relaxation” paragraph: “linear projections of *a* dictionary”
- p. 7, last paragraph of “Architecture and Evaluation” paragraph: switch first sentence to present tense; “total #Params *i*n...”
- p. 8, “Machine Translation” paragraph: “DictFormer obtain*s* more compact”
- p. 8, “Sensitive Study” paragraph: rename to “Sensitivity Study”
- p. 9 , first paragraph: “coefficient size is fixed *to* 60”
- p. 9 , “Ablation” paragraph, first sentence: missing space after period
- p. 9, “We will release code and data...”: Is there data to be released?



**Summary Of The Paper:**

This work proposes a modification of the original Transformer architecture by replacing attention layers and layers in its Feed-Forward Networks across all of its blocks with learned shared dictionaries. The proposed model, called DictFormer, has a smaller number of parameters and uses a smaller amount of computational operations when compared to the original Transformer and some of its variations. When evaluated against these models on popular machine translation, summarization, and language modeling benchmarks, DictFormer achieves comparable or better performance.

**Summary Of The Review:**

The proposed modification to the Transformer architecture is novel and I believe would be interesting for the community but the methodology and motivation could be explained more clearly and provided with more context, including more details on the hyperparameter selection and on how the DictFormer is trained. The experimental results would be even more convincing if confidence intervals are provided.

#### Updates during paper discussion
Based on the author's responses to the reviewers' questions and updates to the manuscript (including clarifying some of their methodology and statements and including confidence intervals in the results section), I've decided to increase my score.

---

> ### Author Response · Authors · 2021-11-19
> **[Part 1] Reply to Reviewer jV7A**
>
> We would like to thank reviewer Jv7A for the thoughtful comments and efforts towards improving our manuscript. Hope our following replies help answer the reviewer's outstanding questions.
>
> ### Question 1:  I might be missing something but why is it stated that the unshared linear projection $\widetilde {W_i^{Q_j}} \approx {W_i}^{Q_j}$. My understanding is that this is not directly optimized for in the model.
>
> It is TRUE. The value of unshared linear projection is NOT approximately equal to the shared weights. We highlight they could have similar representation abilities. For example, the model using the unshared weights and the model using the shared weights have similar model performance.
>
> ### Question 2: Group-wise Shared Dictionary FFN. The motivation behind dividing columns of the dictionary into groups is a bit unclear.  What is meant by “high-quality performance” of the shared dictionary projection?
>
> The “high-quality performance” means an accurate model performance.  The motivation behind dividing columns of the dictionary is that assigning a single scaling coefficient to the whole column may lead to an accuracy decrease, so we propose to assign multiple scaling coefficients to the column of the dictionary. For example, if a column has two groups, each group number will be assigned a scaling coefficient. The overhead of group-wise is that we need more coefficients, which is already considered in our results.
>
> ### Question 3. Have the authors considered using a larger number of dictionary elements m to increase the “flexibility” of the model?
>
> Yes, we study the effects of DictFormer with various dictionary sizes from 80 to 240 and different coefficient sizes from 12 to 84 shown in Figure 6. One can pick up a dictionary size and coefficient size to control the trade-off between accuracy and model size according to the requirements.
>
> ### Question 4. How is the number of groups determined?
>
> The number of groups is determined by our empirical results. In default, the group number can be 1, where the DictFormer can also achieve a compact model without an accuracy decrease shown in Table 3. The group number controls the number of the coefficients. The larger the group number, the more coefficients we have.  One could set the group number as 2 to obtain a better result shown in Table 3 if the hidden size is larger than 1024.
>
> ### Question 5. Since the sparse matrix Z is initialized using values in C, how are coefficients initialized?
>
> All coefficients in our experiments are initialized randomly using the same method as our baselines for a fair comparison.
>
> ### Question 6. Missing confidence intervals.
>
> We added it by providing the average performance across three independent runs.
>
> ### Question 7. How is this work related to work on Sparse Transformers (e.g. [1], [2]) or fixed attention such as [3], [4]?
>
> These four works are related to our DictFormer, and we added them as existing works. Sparse Transformers (e.g. [1], [2]) belong to the first line of research introduced in section 2 of our manuscript. The first line of research is to reduce the transformer computation complexities by redesigning the self-attention mechanism. The difference between these two works with our Dicformer is that their target is not to reduce model size. The fixed attention such as [3], [4] are solely based on positional information and do not require any learnable attention parameters, which is a different and interesting way to improve transformer.
>
> [1] Child R, Gray S, Radford A, Sutskever I. Generating long sequences with sparse transformers. arXiv preprint arXiv:1904.10509. 2019 Apr 23.
>
> [2] Correia, G.M., Niculae, V. and Martins, A.F., 2019. Adaptively sparse transformers. arXiv preprint arXiv:1909.00015.
>
> [3] You, W., Sun, S. and Iyyer, M., 2020. Hard-coded gaussian attention for neural machine translation. arXiv preprint arXiv:2005.00742.
>
> [4] Raganato, A., Scherrer, Y. and Tiedemann, J., 2020. Fixed encoder self-attention patterns in transformer-based machine translation. arXiv preprint arXiv:2002.10260.
>
> ### Question 8. Is it necessary to have the dictionary size less than the embedding size, namely m<d? Would it not be feasible to have a large dictionary (m>d) but keep the number of selected components t small (i.e. t<d) through a sparsity constraint?
>
> It is NOT necessary to have the dictionary size less than the embedding size. It is feasible to have a large dictionary (m>d) but keep the number of selected components. Yes, we study the effects of DictFormer with various dictionary sizes from 80 to 240 and different coefficient sizes from 12 to 84 shown in Figure 6. One can pick up a dictionary size and coefficient size to control the trade-off between accuracy and model size according to the requirements.

---

> > ### Author Response · Authors · 2021-11-19
> > **[Part 2] Reply to Reviewer jV7A**
> >
> > ### Question 9. Have the authors tracked whether all columns of the dictionaries are used in practice?
> >
> > We have verified that all columns of the dictionaries are used. The verification is implemented by tracking if the index matrix covers all the indices of a dictionary. If so, then all columns of the dictionaries are used. Our results show that all columns of the dictionaries are used. This represents the sparse matrix $Z$ does not have a row whose values are all zeros.
> >
> > ### Question 10: Have the authors tracked what percentage of the $t$ coefficients are non-zero on average?
> >
> > Yes. The $t$ coefficients are dense so that 100% values are not zeros. The reviewer may want to ask $Z$ matrix. $Z$ matrix is sparse whose column has $t$ non-zero values on average.
> >
> > ### Question 11: Typos
> >
> > Fixed. We would like to thank again reviewer Jv7A for the thoughtful comments and efforts towards improving our manuscript.

---

> > > ### Comment · Reviewer_jV7A · 2021-11-21
> > > **Considering increasing my score and further suggestions for improvement.**
> > >
> > > Thank you for your detailed response and clarifications! After reading your responses to all the reviews, I'm considering increasing my score.
> > >
> > > Additional Questions
> > > - To clarify Question 10 from above, is it guaranteed that the $Z$ matrix has $t$ non-zero values on average? Is equation 15 actually used in your experiments to force that $t$ of the elements in each column of $Z$ are non-zero while the rest are set to 0?
> > >
> > > Additional suggestions to improve the current (updated) manuscript:
> > > - To avoid confusion as to whether the proposed method learns to predict the weights of a pre-trained Transformer (a concern expressed by reviewer aKni), I would suggest avoiding the word "reconstruct" in the paragraph "Shared-dictionary Attention". For example, in the sentence "This process can reconstruct a $N$-layer weight $W^Q$" on p. 5, I believe a more clear statement would be "This process _defines_ a $N$-layer weight $W^Q$".
> > > - Similarly, following up on my earlier point about the expression "we find ... to meet $\tilde{W_i^{Q_i}} \approx W_i^{Q_i}$" in the paragraph "Shared-dictionary Attention", since parameters in the computation $\tilde{W_i^{Q_i}}$ are trained independently from parameters in the traditional Transformer $W_i^{Q_i}$, I would suggest replacing this statement with something in the spirit of your answer above "We highlight they could have similar representation abilities.".
> > > - In Table 2, the DictFormer Ratio for WIKITEXT103 is 3.8x while in the main text it is stated as 3.7x.
> > > - It would be helpful if Fig. 6 caption includes the task on which the presented models are evaluated.
> > >
> > > I believe addressing the points above would further improve the current version of the work.

---

> > > > ### Author Response · Authors · 2021-11-21
> > > > **Thanks for further valuable suggestions and considering increasing score**
> > > >
> > > > We really appreciate your considerate and valuable suggestions! We have carefully incorporated them into our updated manuscript.  Hope they help reply to your questions and suggestions.
> > > >
> > > > ### Question 1:  To clarify Question 10 from above, is it guaranteed that the $Z$ Matrix has $t$ non-zero values on average? Is equation 15 actually used in your experiments to force that $t$ of the elements in each column of  $Z$ is non-zero while the rest are set to 0?
> > > >
> > > > Exactly, it is guaranteed that the $Z$ Matrix has $t$ non-zero values in each column on average. We added more descriptions for equation 15 in the new version to highlight that one can force that $t$ of the elements in each column of $Z$ are non-zero while the rest are set to 0 by letting $value(\rho)$ is the $t$-th largest value of each column.
> > > >
> > > > ### Question 2:  To avoid confusion as to whether the proposed method learns to predict the weights of a pre-trained Transformer (a concern expressed by reviewer aKni), I would suggest avoiding the word "reconstruct" in the paragraph "Shared-dictionary Attention". For example, in the sentence "This process can reconstruct a $N$-layer weight $W^Q$" on p. 5, I believe a more clear statement would be "This process defines a $N$-layer weight $W^Q$".
> > > >
> > > > Thanks for your considerate suggestions and for even helping us better reply to the questions of reviewer aKni! We have already removed the words or sentences regarding " reconstruction" to avoid confusion. For example, the sentence "linear projections of lookup of $D^A$ with coefficients $C_i^{X}$ can construct $W_i^{X^j}$ " has been updated as "linear projections of lookup of $D^A$ with coefficients $C_i^{X}$ can have similar representation abilities as $W_i^{X^j}$".
> > > >
> > > > ### Question 3: Replacing $\widetilde {W_i^{Q_j}} \approx {W_i}^{Q_j}$” with something in the spirit of your answer.
> > > >
> > > > We really appreciate your considerate suggestions! We replace the sentence “we find proper $D^A$, $I_i^{Q_j}$,$C_i^{Q_j}$ to meet $\widetilde {W_i^{Q_j}} \approx {W_i}^{Q_j}$” with “we find proper $D^A$, $I_i^{Q_j}$,$C_i^{Q_j}$ to meet that $\widetilde {W_i^{Q_j}}$ and ${W_i}^{Q_j}$ have similar representation abilities, e.g., they have matched accuracy.”
> > > >
> > > > ### Question 4: In Table 2, the DictFormer Ratio for WIKITEXT103 is 3.8$\times$ while in the main text it is stated as 3.7$\times$.
> > > >
> > > > DictFormer compresses 3.78$\times$ model size, so 3.8$\times$ is correct. Now we align the table and text with the correct 3.8$\times$ model size compression.
> > > >
> > > > ### Question 5: It would be helpful if Fig. 6 caption includes the task on which the presented models are evaluated.
> > > >
> > > > We added the Fig.6 caption with the task descriptions: (a) and (b) are tested on WMT'14 En-Fr dataset for machine translation tasks.

---

> > > > > ### Comment · Reviewer_jV7A · 2021-11-26
> > > > > **Additional questions**
> > > > >
> > > > > Thank you for your responses! Based on the discussion so far, I've decided to increase my score.

---

> > > > > > ### Author Response · Authors · 2021-11-28
> > > > > > **Thanks for increasing score and your outstanding reviews!**
> > > > > >
> > > > > > Dear Reviewer jV7A,
> > > > > >
> > > > > > Many thanks for your helpful and valuable reviews/discussions!
> > > > > >
> > > > > > Sincerely,
> > > > > >
> > > > > > Paper2101 Authors

---

### Official Review · Reviewer_EmuN · 2021-11-03

**Correctness:** 3
**Technical Novelty And Significance:** 3
**Empirical Novelty And Significance:** 3
**Recommendation:** 6
**Confidence:** 2

**Main Review:**

The paper contains a lot of substance, but it is very dense and hard to follow. The core “dictionary” technique isn’t really explained at a high level before the paper plunges into the details. It seems to be something like the approach in [1] but it’s difficult to be sure (I gave up on section 3 after a while). The results in section 5 are very impressive, but some intuition about why a compressed approach like this could beat a much larger baseline on large data settings really need to be provided.

[1] Kaiser, Lukasz, et al. "Fast decoding in sequence models using discrete latent variables." International Conference on Machine Learning. PMLR, 2018.

Details:
- The “first line of research”: would be good to add a word or two saying how these papers reduce computational complexity.
- Figure 1 is really great, but you should say where these stats come from.
- Figures 2 and 3: captions crash into text.
- This is hard to understand: “	In this paper, the #Params omit word embedding size that would highly dependent on the sentence length and would significantly differ for various tasks. The total #Params in this paper includes the model size of word embedding.”
- It’s difficult to align table 3 with figure 5. You should include a line corresponding to the point in figure 5 with highest BLEU (higher than anything that appears in table 3).


**Summary Of The Paper:**

This paper describes a technique for reducing the size and computation of a Transformer model by projecting and factoring weight matrices. Experiments on MT, summarization, and language modeling show improved results over competing techniques, and even over standard Transformers, despite using significantly fewer parameters and less computation.


**Summary Of The Review:**

Potentially a great paper, but if so it deserves to be much better explained.

---

> ### Author Response · Authors · 2021-11-19
> **Reply to Reviewer EmuN**
>
> We would like to thank reviewer EmuN for the thoughtful comments and efforts towards improving our manuscript. Hope our following replies help answer the reviewer's outstanding questions.
>
> ### Question 1: Intuition and the high-level idea of the dictionary should be explained more, e.g., why dictionary works?
>
> The dictionary sharing is motivated by the problems of existing Transformer weights sharing works: Directly sharing weights across layers in the transformer either decreases the model performance [1] or prolongs the computational latency by stacking more layers [2].  This issue indicates that not all weights can be shared across layers which means that partial information in weights can be shared, and the others should be kept distinct.  In order to solve the issues of weights sharing.   We define these partial weights that can be shared as dictionaries and use distinct and small indexes and coefficients to represent weights information that cannot be shared.
>
> Our work is different from the approach in [3] that uses improved semantic hashing on latent features to speed up the neural machine translation task.  As far as the authors know, the approach in [1] does not focus on reducing the model size of transformers. The improved semantic hashing in [2] is fundamentally different from our dictionary, index, and coefficients. For example, the hash data structure is not used in our method. We have already added [3] to the existing works.
>
> [1] Machel Reid, Edison Marrese-Taylor, & Yutaka Matsuo (2021). Subformer: Exploring Weight Sharing for Parameter Efficiency in Generative Transformers. In Findings of the Association for Computational Linguistics: EMNLP 2021. Association for Computational Linguistics.
>
> [2] Mostafa Dehghani and Stephan Gouws and Oriol Vinyals and Jakob Uszkoreit and Lukasz Kaiser (2018). Universal Transformers. CoRR, abs/1807.03819.
>
> [3]Kaiser, L., Bengio, S., Roy, A., Vaswani, A., Parmar, N., Uszkoreit, J., & Shazeer, N. (2018). Fast Decoding in Sequence Models Using Discrete Latent Variables. In Proceedings of the 35th International Conference on Machine Learning (pp. 2390–2399). PMLR.
>
> ### Question 2:  It would be good to add a word or two to explain how the first line of research reduces computational complexity?
>
> The first line of research is to reduce the transformer computation complexities by redesigning the self-attention mechanisms including informer[1], sparse attention [2].
>
> [1] Haoyi Zhou and Shanghang Zhang and Jieqi Peng and Shuai Zhang and Jianxin Li and Hui    Xiong and Wancai Zhang (2020). Informer: Beyond Efficient Transformer for Long Sequence Time-Series Forecasting. CoRR, abs/2012.07436.
>
> [2] Child R, Gray S, Radford A, Sutskever I. Generating long sequences with sparse transformers. arXiv preprint arXiv:1904.10509. 2019 Apr 23.
>
> ### Question 3: Figure 1 is really great, but you should say where these stats come from.
>
> We added more explanations to describe Figure 1 in the new manuscript. For example, the results of existing works in Figure 1 come from existing implementations and \#Params does not include embedding parameters, e.g., Transformer has 44M \#Params and achieves 27.3 BLUE on WMT-14 En-De dataset [1][2]. Parameters can be derived and verified by transformer [1] or existing works DeLighT [2].
>
> [1] Vaswani, Ashish et al. "Attention is All you Need." Advances in Neural Information Processing Systems. Curran Associates, Inc.,
>
> [2] ] Sachin Mehta, Marjan Ghazvininejad, Srinivasan Iyer, Luke Zettlemoyer, & Hannaneh Hajishirzi (2021). DeLighT: Deep and Light-weight Transformer. In International Conference on Learning Representations (ICLR).
>
> ### Question 4: Figures 2 and 3: captions crash into text.
>
> Fixed in the current version.
>
> ### Question 5: This is hard to understand: “ In this paper, the #Params omit word embedding size that would highly dependent on the sentence length and would significantly differ for various tasks. The total #Params includes the model size of word embedding.”
>
> We removed the above confusing expressions by adding a clear one: “In this paper, \#Params means parameter numbers of the transformer without including embedding layers;  \#TParams shows the total parameter numbers of model and embedding layers.”
>
> ### Question 6: It’s difficult to align table 3 with figure 5. You should include a line corresponding to the point in figure 5 with the highest BLEU (higher than anything that appears in table 3).
>
> After we calculate and update the average BLEU across three independent runs in table 3, it is aligned between table 3 and figure 5.

---

> > ### Author Response · Authors · 2021-11-22
> > **Available for Discussion**
> >
> > We hope our responses have addressed the reviewer's concerns, but if not we are available/open to address any outstanding issues.

---

> > > ### Comment · Reviewer_EmuN · 2021-11-26
> > > **More positive assessment**
> > >
> > > Having read the other reviews and the author responses, I have a better understanding of the contribution. I think there’s enough substance, novelty, and good experimental results here to warrant acceptance, and I’ve revised my score accordingly.
> > >
> > > The main weakness is still the lack of a good high-level explanation. In section 3, “Overview”, instead of starting with an overview of the computational advantages, I suggest you start with an overview of the idea. Explain what the dictionary is, what components are specific to each layer (indices, coefficients), and how these work together.

---

> > > > ### Author Response · Authors · 2021-11-28
> > > > **Thanks for increasing score and positive suggestions**
> > > >
> > > > Dear Reviewer EmuN,
> > > >
> > > > We appreciate very much your increased score and positive suggestions! We make sure that we will follow your suggestions and add more high-level definitions and descriptions regarding dictionary ideas!
> > > >
> > > > Sincerely,
> > > >
> > > > Paper2101 Authors

---

### Official Review · Reviewer_aKni · 2021-11-03

**Correctness:** 3
**Technical Novelty And Significance:** 2
**Empirical Novelty And Significance:** 2
**Recommendation:** 6
**Confidence:** 5

**Main Review:**

The idea of the proposed method is interesting, but there are a few concerns in terms of the presentation.
Therefore, it is hard to judge whether this paper has enough contribution for publishing as the conference paper.

The following are my concerns in the current version.

### 1, Technical novelty
* The idea of the proposed method is interesting and might be effective.
However, the idea itself of sharing the parameter is not very innovative.
I think that sharing parameters for compressing DNNs is a standard technique nowadays.
Therefore, the authors need to clarify the contributions of the proposed method, such as the unique properties that previous similar compression methods cannot achieve.
Currently,  I do not find any strong properties in the proposed method.

* If my understanding is correct, the proposed method is a reconstruction method.
Therefore, we need a trained model for applying the proposed method.
This means the proposed method requires additional computation.
I do not fully understand why this paper compares the computational cost with the standard Transformer.

### 2, Notation and equation
* The notations are incredibly messy and hard to understand.
The authors need to make notations much simpler for better understanding to readers.

### 3, L1 constraint
* If my understanding is correct, the relaxed L1 constraint does not guarantee to find the solution that satisfies the threshold of non-zero factors. This paper seems not to explain the way if such a situation occurs in the solution. ​

### 4, Typo or misconfiguration?
* In Table 1, it says the results for WMT De-En and WMT Fr-En.
However, at the beginning of Section 4, the experiments are conducted on WMT "En-De" and "En-Fr," which are not "De-En" and "Fr-En."


### 5, Confirmation of model sizes
* According to the original Transformer paper [1], the numbers of parameters of Transformer (base) and (large) are 64M and 213M, respectively.
However, in the experiments, the model size of the baseline Transformer is 3.6M (as shown in Table 1) for WMT En-De.
Moreover, I checked the previous paper, such as the "Lite Transfomer" paper (Wu et al., 2020) and the "Pay less attention" paper (Wu et al., 2019).
However, I could not find the precise experimental settings used in this paper.
I recommend clearly showing the model configurations and hyper-parameter settings for keeping reproducibility.
Otherwise, the reproducibility of the proposed method may not be sufficient.

[1] Vaswani et al., Attention Is All You Need, In Proc. of NIPS-2017.




### 6, Inconsistent results in Table 1 and 3
* I thought that the ablation study of Table 3 is based on the results (settings) of Table 1.
However, the numbers of parameters shown in Tables 1 and 3 differ entirely, so I do not understand the meaning of the ablation study in Table 3.
Please confirm it and clarify the configuration difference between Tables 1 and 3.
Moreover, explain the results of the baseline Transformer and the proposed method corresponding to the ablation results in Table 3.

* Additionally, it seems that there is no description about what the "Improved-Embedded" is shown in Table 3.
If I miss the description, please let me know. If the paper lacks explanation, this can be an apparent problem for this paper in terms of completeness.














**Summary Of The Paper:**

This paper proposes a compression method for Transformer-based encoder-decoder or language models.
The key idea of the proposed method is to decompose the standard parameters into a much smaller shared parameter matrix and independent parameters for each original matrix.
Then, the method can approximately recover the original Transformer models by simple additions and multiplications.

The experiments are conducted on three MT tasks, one summarization task, and one language modeling task.
Experimental results show that the proposed method seems to reduce model sizes and computations successfully while preventing considerable performance degradation (in some cases, the proposed method appears to improve the performance).

**Summary Of The Review:**

The idea of the proposed method is interesting and might be effective.
However, the idea itself of sharing the parameter is not very innovative and rather incremental.
Experimental settings are ambiguous and seems to use very weak settings.

---

> ### Author Response · Authors · 2021-11-19
> **Reply to Reviewer aKni**
>
> We would like to thank reviewer aKni for the thoughtful comments and efforts towards improving our manuscript. Hope our following replies help answer the reviewer's outstanding questions.
>
> ### Question 1: Technical novelty (Parameter Sharing is not innovative).
>
> Our proposed DictFormer has multiple novelties including Dictionary-Index-Coefficient architecture for light-weight transformer, shared dictionaries in attention and FFN, group-wise dictionaries, and their training implementations.
>
> We totally agree that “sharing parameters for compression DNNs is a standard technique nowadays”, but we argue that we use DICTIONARY sharing instead of regular previous weights sharing. Section 2 in our paper states that direct weights sharing in transformers suffer from severe issues: Low accuracy or heavy computations.
>
> DICTIONARY sharing could reduce computation tasks, but weights sharing cannot reduce computations due to the proposed smaller dictionary architecture (dictionary size is less than weight size) and shared dictionaries in query, key, and values (the multiplication between hidden state and shared dictionaries can be reused that is showed in equation 5 ). Cross-layer DICTIONARY sharing can still enable unique weight for each layer since the index and coefficients are not shared. In contrast, cross-layer sharing cannot enable unique weight in each layer since various layers use the same weight.
>
>
> ### Question 2: If my understanding is correct, the proposed method is a reconstruction method. Therefore, we need a trained model…
>
> The proposed method is NOT a reconstruction method. The proposed DictFormer is a new lightweight architecture with shared dictionaries so that we could use DictFormer to replace baseline transformer for smaller model sizes, especially when deploying models on mobile devices. Therefore, a trained/pre-trained model is not necessarily required. We could randomly initialize dictionaries, indices, coefficients, and train them from scratch as the transformer does.
>
> ### Question 3: WMT De-En and WMT Fr-En.
>
> It is a typo. We fixed it by changing them into WMT En-De and WMT En-Fr.
>
>
> ### Question 4: Transformer Model sizes.
>
> The parameter numbers of the original transformer are ~64M that is added into Table 3 in our new version. For mobile settings, we follow lite-transformer [We et al., 2020] that creates a transformer with 2.8M parameters by scaling down the hidden model size of original transformer from 512 to 128, the feed-forward size from 2048 to 512.  (This does not contain parameters in embedding. ) , and we exactly use this setting in Table 1. So the baseline transformer in table 1 is 2.8M. And 3.6M is a typo. The results of other existing works in Table 1 are obtained by scaling down the hidden size using their own implementations for a fair comparison.
>
>
> ### Question 5: Inconsistent results in Tables 1 and 3.
>
> The ablation study of Table 3 is NOT based on the results of Table 1. We use Table 1 to compare our DictFormer with the state-of-the-art lite-transformer [1]. However, we use Table 3 to show the ablation study of our multiple techniques that are based on the original transformers (64M total parameters and 44M transformer parameters). We appreciate the reviewer’s advice to add the original transformer in Table 3.  We added it to the current manuscript and showed that the improvement of our technique over the baseline transformer.  In addition, we added an explanation on “Improved-Embedded”  that represents that the embedding layers of DictFormer are further compressed by using an existing method, DeLighT [2], so that the \#TParams is reduced to 5.1M from 7.1M.
>
>
> [1] Zhanghao Wu*, Zhijian Liu*, Ji Lin, Yujun Lin, & Song Han (2020). Lite Transformer with Long-Short Range Attention. In International Conference on Learning Representations (ICLR).
>
> [2] Sachin Mehta, Marjan Ghazvininejad, Srinivasan Iyer, Luke Zettlemoyer, & Hannaneh Hajishirzi (2021). DeLighT: Deep and Light-weight Transformer. In International Conference on Learning Representations (ICLR).

---

> > ### Author Response · Authors · 2021-11-22
> > **[Part 2] Reply to Reviewer aKni**
> >
> > ### Question 6: L1 constraint. If my understanding is correct, the relaxed L1 constraint does not guarantee finding the solution that satisfies the threshold of non-zero factors. This paper seems not to explain the way if such a situation occurs in the solution.
> >
> > The L1 constraint CAN guarantee finding the solution that satisfies the threshold of non-zero factors. This is because Equation 15 can force $t$ non-zeros on average in each column of the $Z$ matrix by keeping the top $t$-largest values and setting the other smaller values as zeros.

---

> > > ### Comment · Reviewer_aKni · 2021-11-29
> > > **misleading derivation of L1-norm regularization**
> > >
> > > Thank you for the response to my question.
> > > I somehow missed Eq 15 when I wrote the first review.
> > >
> > > However, I have to point out that Eq 15 is not the same operation as L1 regularization in theory.
> > > Therefore, the authors need to note this fact to prevent misunderstanding among readers.
> > >
> > > Please refer to the paper,
> > > [Duchi+, ICML'08] "Efficient Projections onto the L1-Ball for Learning in High Dimensions"
> > >
> > > It is true that Eq. 15 looks similar to the projection operation onto the L1-norm ball, but not the same.
> > > The crucial difference is that every value should be decreased by the regularization constant.
> > > Moreover, the regularization constant should be the "constant," not be an adaptively changing value like $\rho$.
> > > Based on this fact, the derivation of Eq 15 by supporting L1-norm regularization is wrong.
> > > Therefore, the proposed method uses the ad-hoc L0-norm-like operation that is not supported by any theoretical interpretation.
> > >
> > >
> > > Even if we accept Eq 15, we can easily find several simple corner cases that do not satisfy the condition.
> > > For example, what would we do if we obtain less than $t^A$ non-zero elements, or every element has an identical value before applying Eq 15?
> > > In such a case, the method cannot guarantee the number of non-zero values in the solution.

---

> > > > ### Author Response · Authors · 2021-11-29
> > > > **Reply to reviewer aKni regarding Eq.14 and Eq.15**
> > > >
> > > > Thanks for your considerate response. It seems that there is a misinterpretation. We would like to highlight that Eq. 15 is NOT a L1-norm regularization, instead Eq. 14 ( $\frac{\delta(L + \lambda || Z||_{l_1})}{\delta Z } = \frac{\delta L}{ \delta Z} + \lambda sign(Z)$) is a L1-norm regularization where we regularize $Z$ to increase sparsity and $\frac{\delta L}{ \delta Z}$ is the gradient that is computed through a standard back-propagation. $\lambda$ is a hyperparameter that adjusts the trade-off between the transformer loss function and the L1 regularizer.   Instead, Eq 15 is a threshold function over the values in $Z$ during training. We also backpropagate through this threshold function to compute the gradients with respect to $Z$. The derivative of the threshold function is 1 everywhere except at $|x|<= value(\rho)$. When $|x|<= value(\rho)$, the derivative is 0. Hence, if any of the entries of $Z$ becomes 0 at some iteration, they stay 0 forever. Using the threshold function, we could make sure that $Z$ AT MOST has $t^A$ non-zeros. It is true that in theory, we have a chance to obtain less than $t^A$ non-zero elements if the original $Z$ already has less than $t^A$ non-zeros. If this case happens, our method still works since our method keeps all the non-zeros naturally.
> > > >
> > > > We appreciate it very much if the reviewer let us know if our reply helped resolve this question. We will add the resolved explanation to the next-version manuscript to avoid misinterpretation.

---

> > > > > ### Comment · Reviewer_aKni · 2021-11-29
> > > > > **Thank you for the prompt response**
> > > > >
> > > > > Thank you for the prompt response.
> > > > >
> > > > > I know that Eq 14 is the gradient with the L1 regularizer (I used to work on the L1 regularized sparse modeling).
> > > > > I guessed that the authors introduced Eq 15 because the authors knew the equivalent relationship between L1-norm projection and L1-norm regularization described in the paper [Duchi+, ICML'08].
> > > > > However, if the authors say that Eq 15 is not directly related to the L1 regularizer, then my next question is what is the theoretical justification of using Eq 15, and why did Eq 15 suddenly appear (relation between Eqs 14 and 15)?
> > > > > I would like to know whether Eq 15 has clear theoretical support or not.

---

> > > > > > ### Author Response · Authors · 2021-11-29
> > > > > > **Reply to Reviewer aKni about Eq.15**
> > > > > >
> > > > > > Thank you for the prompt and thoughtful feedback.
> > > > > >
> > > > > > ### Question: What is the theoretical justification of using Eq 15, and why did Eq 15 suddenly appear (relation between Eqs 14 and 15)?
> > > > > >
> > > > > > Since the reviewer used to work on the L1 regularized sparse modeling, I believe you know that only using Eq 14 can only reduce the L1 norm size of parameter $Z$. In other words,  Eq.14 can increase the number of small values in $Z$ but cannot make sure the small values are zeros. Eq.15 is used to further prune the small values to zeros. Therefore, we believe that it is natural to use Eq 15 after Eq 14. The theoretical justification of using Eqs 14 and 15 is inspired by the previous pruning method [1] [2], where L1 or L2 regularizers are used to foster sparsity, and then unimported small values are pruned.
> > > > > >
> > > > > > [1] Wen, W., Wu, C., Wang, Y., Chen, Y., & Li, H. (2016). Learning structured sparsity in deep neural networks. Advances in neural information processing systems, 29, 2074–2082.
> > > > > >
> > > > > > [2]  Li, H., Kadav, A., Durdanovic, I., Samet, H., & Graf, H. (2016). Pruning filters for efficient convnets. arXiv preprint arXiv:1608.08710.

---

> > > > > > > ### Comment · Reviewer_aKni · 2021-11-30
> > > > > > > **Reply about the derivation of Eq 15**
> > > > > > >
> > > > > > > * "Eq.14 can increase the number of small values in  $Z$ but cannot make sure the small values are zeros."
> > > > > > >
> > > > > > > I respectfully disagree with the above response; this is not true.
> > > > > > > There are several well-studied techniques that can guarantee a sparse solution in online learning settings (of course including mini-batch learning with a straightforward extension), such as [3,4,5] and their variants.
> > > > > > >
> > > > > > > [3] Duchi+, ICML'08, "Efficient Projections onto the L1-Ball for Learning in High Dimensions"
> > > > > > > [4] Tsuruoka+, ACL'09, "Stochastic Gradient Descent Training for L1-regularized Log-linear Models with Cumulative Penalty"
> > > > > > > [5] Duchi+, NIPS'09, "Efficient Learning using Forward-Backward Splitting"
> > > > > > >
> > > > > > >
> > > > > > > * "The theoretical justification of using Eqs 14 and 15 is inspired by the previous pruning method [1] [2], where L1 or L2 regularizers are used to foster sparsity, and then unimported small values are pruned."
> > > > > > >
> > > > > > > I am afraid that the above statement cannot be a formal theoretical justification for deriving Eq15 from Eq 14.
> > > > > > >
> > > > > > > Moreover, I have skimmed the papers [1] and [2].
> > > > > > > The paper [1] seems their method is a standard $L_{2,1}$-regularization method, not a pruning method like Eq15. Therefore, the authors' interpretation is somewhat wrong.
> > > > > > >
> > > > > > > The authors' proposal of using the pruning method is closely similar to [2].
> > > > > > > In fact, the paper [2] says, "we use l1-norm to select unimportant filters and physically prune them."
> > > > > > > Then, the next question is why the authors did not refer to their paper even the authors say, "Eqs 14 and 15 is inspired by the previous pruning method [1] [2]," in their response?
> > > > > > > I believe every researcher should respectfully refer to previous studies without intentional omission.
> > > > > > >
> > > > > > > Note that I am not saying that using Eq 15 is wrong, but asking how this pruning operation can be derived.
> > > > > > > The current explanation seems to say that Eq 15 was derived from L1 regularization (equivalent to L1 norm constraint).
> > > > > > > However, it is misleading and wrong, so that I recommend revising the explanation of this part by referring to the previous pruning method [2].

---

> > > > > > > > ### Author Response · Authors · 2021-11-30
> > > > > > > > **Revised Paper and More Explanations**
> > > > > > > >
> > > > > > > > ### 1. We have cited [2] in the current manuscript (Unfortunately, the current manuscript cannot be submitted).
> > > > > > > >
> > > > > > > > Currently, the sparsity strategy methods [1][2] that depend on L1 norm regularization and pruning (clipping small values) are widely used and adapted for research purposes. Our methods are not exactly the same as the methods in [2].  We adapt the regular methods based on L1 norm regularization and small values clipping to obtain a sparsity matrix for our dictionary sharing scheme. We are not motivated to intentionally omit the pruning methods.  However, it is also a valuable suggestion to cite the work [2] to help readability and reproducibility.
> > > > > > > >
> > > > > > > > ### 2. We have revised the explanation to Equation 15 in the current manuscript.
> > > > > > > >
> > > > > > > > We have followed your suggestions and revised the explanation to Eqs 13, 14, 15 to avoid misleading points. (Unfortunately, the current manuscript cannot be submitted).
> > > > > > > >
> > > > > > > > ### 3. Besides the manuscript changes mentioned above, we add more explanations to our statement "Eq.14" as follows:
> > > > > > > >
> > > > > > > > We totally agree with the reviewer's comment: *There are several well-studied techniques that can guarantee a sparse solution[3][4][5]*. We would like to highlight that OUR Eq. 14 without Eq. 15 cannot guarantee a sparse solution. Eq. 15 is used to clip the small values into zero according to their absolute value. Section 3.1 in [4] can potentially help verify our claims. The SGD-L1 (Naive) in [4] shares the same theory as our Eq 14, and the paper [4] states in section 3.1 that *"it, i.e., SGD-L1 (Naive) does not produce a compact model, i.e. most of the weights of the features do not become zero as a result of training. Note that the weight of a feature does not become zero unless it happens to fall on zero exactly, which rarely happens in practice"*, which shows that only using Eq 14 usually cannot make sure the values are exactly zeros although some values are smaller than 1e-6. This point is also verified by our experiments.  Moreover, in order to solve this issue, section 3.1 in [4] also states that *"In other words, the weight is clipped when it crosses zero."*, which is exactly what our Eq.15 does.
> > > > > > > >
> > > > > > > > The *SGDL1 (Clipping)* method in section 3.1 [4] is like the combination of our Eq. 14 and Eq. 15.  Although this method may be not perfect, it has several advantages: *The obvious advantage of using this method is that we can expect many of the weights of the features to become zero during training. Another merit is that it allows us to perform the application of the L1 penalty in a lazy fashion, so that we do not need to update the weights of the features that are not used in the current sample, which leads to much faster training when the dimension of the feature space is large.*
> > > > > > > >
> > > > > > > > We would appreciate it very much if the reviewer could let us know if our explanations help solve the concerns.
> > > > > > > >
> > > > > > > > [1] Wen, W., Wu, C., Wang, Y., Chen, Y., & Li, H. (2016). Learning structured sparsity in deep neural networks. Advances in neural information processing systems, 29, 2074–2082.
> > > > > > > >
> > > > > > > > [2] Li, H., Kadav, A., Durdanovic, I., Samet, H., & Graf, H. (2016). Pruning filters for efficient convnets. arXiv preprint arXiv:1608.08710.
> > > > > > > >
> > > > > > > > [3] Duchi+, ICML'08, "Efficient Projections onto the L1-Ball for Learning in High Dimensions"
> > > > > > > >
> > > > > > > > [4] Tsuruoka+, ACL'09, "Stochastic Gradient Descent Training for L1-regularized Log-linear Models with Cumulative Penalty"

---

> > ### Comment · Reviewer_aKni · 2021-11-29
> > **Thank you for your response**
> >
> > Thank you for the detailed response to my questions and concerns.
> >
> > Question 1:
> >
> > In my view, I felt that dictionary sharing and standard parameter sharing (what the authors claim differ) are in the same group of parameter compression methods
> > since the basic structure of sharing the parameter is the same.
> > However, after I read the rebuttal, I also understand the authors' claim that dictionary sharing has some contributions in terms of a technical novelty.
> >
> >
> > Question 2:
> >
> > I am sorry for my misunderstanding.
> > Now, I understand that the experimental setting used in this paper is a fair condition.
> >
> >
> > Question 3:
> >
> > Thank you for correcting the errors of critical information for reproducibility.
> >
> >
> > Question 4
> >
> > Thank you for the detailed explanation.
> > I usually use Transformer base and large (or even larger) settings in my study and have never used a Lite-Transformer setting. Therefore, I am not familiar with such a setting.
> > For clarification, let me confirm one more thing about the settings;
> > The author mentioned, "For mobile settings, we follow lite-transformer [We et al., 2020] that creates a transformer with 2.8M parameters...".
> > However, the #params of Lite Transformer in Table 1 were 2.3M and 1.8M for IWSLT'14 De-En and WMT'14 En-De, respectively, which are not 2.8M.
> > I am still confused about this inconsistency... I guess I may misunderstand something.
> > However, please clarify this point since this is important for reproducing the experiments in future studies.
> >
> >
> > Question 5:
> >
> > I understand that the authors used the different model configurations for the main results (Table 1) and ablation study (Table 3).
> > However,  I am still wondering why the authors need to use inconsistent configurations.
> > I guess that suddenly using a new setting in the analysis part (ablation study) without an explanation is a bit confusing for readers.
> > To alleviate this confusion,
> > it would be much better to show the comparison on standard transformer setting for main results (Table 1) in addition to the lite-transformer settings
> > since I believe there is no particular reason that the authors cannot run their experiments in the standard transformer setting.
> >
> > Thank you for explaining the “Improved-Embedded” setting and also other settings used in Table 3.
> >
> >
> >
> > Overall, the authors fixed and added several important pieces of information (Q3: "It is a typo. We fixed it by changing it into WMT En-De and WMT En-Fr." Q4: "...And 3.6M is a typo." Q5: adding the explanation of "Improved-Embedded," and so on).
> > Based on these facts, I believe that my assessment for the first version was not so harsh.
> > Then, the current version becomes much better by the completion of missing and error information.
> > Therefore, I will consider increasing the overall assessment.
> > Besides, it would be great if the authors can answer the additional questions/confirmations in Q4 and Q5.

---

> > > ### Author Response · Authors · 2021-11-29
> > > **Thanks for considering increasing the overall assessment**
> > >
> > > Question 1: Thanks for the reviewer’s understanding that dictionary sharing has some contributions in terms of a technical novelty. Also, we would highlight again that our results show that sharing only dictionaries reduces accuracy loss, compared to previous parameter sharing.
> > >
> > > Question 2 and question 3: Thanks for the reviewer’s thoughtful comments and efforts towards improving our manuscript.
> > >
> > > ### Question 4:  Model size and lite-transformer mobile settings.
> > >
> > > “For mobile settings, we follow lite-transformer [We et al., 2020] that creates a transformer with 2.8M parameters by scaling down the hidden model size of the original transformer from 512 to 128, the feed-forward size from 2048 to 512.” This 2.8M model size is for the transformer settings in table 1, not for the lite-transformer in table 1. For Lite-transformer with attention branch and convolution branch in table 1, we did not directly use its 2.8M model (hidden model size of 160 and feed-forward size 160) reported in its paper, instead, we scaled down the model into 2.3M (hidden model size of 140 and feed-forward size 140)  and 1.8M (hidden model size of 128 and feed-forward size 128) to match the accuracy of transformer with the size of 2.8M. The reason why we did this scaling-down is to fairly compare the model size of baseline transformer and existing lightweight works under a similar accuracy constraint. In other words, given the 27.8 BLEU score, we want to use table 1 to show which model can achieve a smaller model size. Compared to the current 2.3M lite-transformer, the 2.8M lite-transformer should be a weak baseline.
> > >
> > > To avoid the confusing point, we will add the 2.8M lite-transformer item in table 1 in the next-version manuscript for more straightforward reproducibility.
> > >
> > > Why lite-transformer settings?
> > >
> > > I understand that the reviewer usually uses large transformer settings. But the compact transformer settings are also meaningful and useful for real-world mobile devices. Multiple prior works like lite-transformer [We et al., ICLR 2020] formalize the deployment of lightweight transformers on mobile devices by defining the mobile settings regarding the amount of computation and parameter numbers. Particularly, given the ARM Cortex-A72 mobile CPU with computation performance of 48G FLOPS, the computation constraint for machine translation should be under 500M Mult-Adds and the parameter number should be around 10M with a sequence of 30 tokens that are the general length for machine translation. Therefore, the lite-transformer [We et al., ICLR 2020] creates a tiny transformer with a size of 2.8M.
> > >
> > > ### Question 5:  Why do the authors need to use different configurations in Table 1 and Table 3?
> > >
> > > The main reason is that we would like to show that DictFormer is scalable on the tiny transformer and regular transformer. We use Table 1 to compare our DictFormer with the state-of-the-art lite-transformer, like [We et al., ICLR 2020] to show how small the transformer model size we can achieve for mobile settings. Then we use Figure 5 and Figure 6 to show that we can scale up the model size and achieve a larger model size. Hence, Figure 5 and Figure 6 cover both results in Table 1 (small model size) and Table 3 (regular model size). Then our Table 3 further shows the model size comparison between the regular transformer and our DictFormer. Besides, Table 3 shows the ablation study of our techniques. Therefore, we believe our results are consistent.  Currently results already show that we can run our experiments in the standard transformer setting and mobile settings.

---

> ### Author Response · Authors · 2021-11-22
> **Available for Discussion**
>
> We hope our responses have addressed the reviewer's concerns, but if not we are available/open to address any outstanding issues.

---

> > ### Author Response · Authors · 2021-11-30
> > **Follow-ups Before discussion channel is Closed**
> >
> > Dear Reviewer aKni,
> >
> > Considering the discussion channel is going to be closed, we sincerely ask if there still are any concerns that are not solved? If so, could you list your follow-up questions? We still have time to discuss them. Thanks!
> >
> > Bests,
> > Paper2101 authors

---

> ### Comment · Area_Chair_ihT1 · 2021-11-26
> **reminder**
>
> Dear reviewer, it seems that the authors gave a substantial response to the initial review. Can you check if they addressed your concerns?

---

### Decision · Program_Chairs · 2022-01-20

**Decision:**

Accept (Poster)

**Comment:**

DictFormer is a method to reduce the redundancy in transformers so they can deployed on edge devices. In the method, a shared dictionary across layers and unshared coefficients are used in place of weight multiplications. The author proposed a l1 relaxation to train the non-differentiable objective  to achieve both higher performance and lower parameter counts.

All reviewers ended up giving the paper a score of 6 after increasing their scores during discussions. While the results are strong (better performance at much lower parameter counts), the paper is not clearly written. Several reviewers noted that the paper is difficult to understand and has a few unresolved points. For example, the method also ended up performing better than the base transformer model that DictFormer is supposed to compress. There seems to be a lack of understanding about what part of the model delivered the improvements. One reviewer said that this is potentially a great paper that deserves to be better explained. The basic concept of sharing a dictionary across layers should be simple enough to explain well and deserve a better explanation than eq 5.

The authors promise to release the code, which would be necessary for a full dissemination of this work. I recommend accept.